# Snow driven uncertainty in CryoSat-2 derived Antarctic sea ice thickness -
insights from McMurdo Sound
Daniel Price[1], Iman Soltanzadeh[2] & Wolfgang Rack[1], Ethan Dale[3]
[1]Gateway Antarctica, University of Canterbury, Private Bag 4800, Christchurch, New Zealand
[2]Met Service, 30 Salamanca Road, Kelburn, Wellington, 6012, New Zealand
[3]Department of Physics and Astronomy, University of Canterbury, Christchurch, New Zealand
*Correspondence to*: Daniel Price (daniel.price@canterbury.ac.nz)
**Abstract.** Knowledge of the snow depth distribution on Antarctic sea ice is poor but is critical to
obtaining sea ice thickness from satellite altimetry measurements of freeboard. We examine the
usefulness of various snow products to provide snow depth information over Antarctic fast ice in
McMurdo Sound with a focus on a novel approach using a high-resolution numerical snow
accumulation model (SnowModel). We compare this model to results from ECMWF ERA-Interim
precipitation, EOS Aqua AMSR-E passive microwave snow depths and *in situ* measurements at the end
of the sea ice growth season in 2011. The fast ice was segmented into three areas by fastening date and
the onset of snow accumulation was calibrated to these dates. SnowModel captures the spatial snow
distribution gradient in McMurdo Sound and falls within 2 cm snow water equivalent (swe) of *in situ*
measurements across the entire study area. However, it exhibits deviations of 5 cm swe from these
measurements in the east where the effect of local topographic features has caused an overestimate of
snow depth in the model. AMSR-E provides swe values half that of SnowModel for the majority of the
sea ice growth season. The coarser resolution ERA-Interim, produces a very high mean swe value 20
cm higher than *in situ* measurements. These various snow datasets and *in situ* information are used to
infer sea ice thickness in combination with CryoSat-2 (CS-2) freeboard data. CS-2 is capable of
capturing the seasonal trend of sea ice freeboard growth but thickness results are highly dependent on
what interface the retracked CS-2 height is assumed to represent. Because of this ambiguity we vary
the proportion of ice and snow that represents freeboard – a mathematical alteration of the radar
penetration into the snow cover and assess this uncertainty in McMurdo Sound. The range in sea ice
thickness uncertainty within these bounds, as means of the entire growth season are 1.08 m, 4.94 m and
1.03 m for SnowModel, ERA-Interim and AMSR-E respectively. Using an interpolated *in situ* snow
dataset we find the best agreement between CS-2 derived and *in situ* thickness when this interface is
assumed to be 0.07 m below the snow surface.
## 1 Introduction
The knowledge of Antarctic sea ice extent, area, drift and roughness have been greatly
improved over the last forty years, principally supported by satellite remote sensing.
Nevertheless, many knowledge gaps remain which restrict our ability to better understand the
Antarctic sea ice system. A foremost concern is inadequate data for the snow depth distribution
on Antarctic sea ice (Pope et al., 2016) as the presence of snow has many important
implications for the sea ice cover (Massom et al., 2001, Wu et al., 1999, Fichefet and Maqueda,
1999). The thermal conductivity of snow is almost an order of magnitude less than sea ice
(Maykut and Untersteiner, 1971) and as snow accumulates, it reduces the conductive heat flux
from the ocean to the atmosphere, slowing growth rates, but also leads to thickening of the ice
cover through snow-ice formation (Maksym and Markus, 2008). Snow significantly increases
the albedo of the sea ice cover and in the austral spring and summer snow melt s responsible
for fresh water input to the Southern Ocean (Massom et al., 2001). Perhaps most crucially from
a satellite observation perspective, our inability to accurately monitor its depth and distribution
causes large uncertainty when estimating sea ice thickness. Sea ice thickness measurements as

inferred via satellite freeboard estimates (Schwegmann et al., 2016, Kurtz and Markus, 2012, Giles et al., 2008) currently present the the best opportunity to establish yet unpublished datasets on decadal trends in Antarctic sea ice volume. Without improved snow depth measurements, it is impossible to discern meaningful trends in Antarctic sea ice thickness. Errors are introduced to thickness estimates via the snow cover for two principal reasons:

1. Snow depth information is inaccurate/not available and therefore the ratio of ice and snow above the waterline is poorly quantified or unknown.
2. Uncertainty about what surface the retracking point on the radar waveform actually represents between the ice freeboard and snow freeboard. This initial measurement is commonly referred to as radar freeboard.

The uncertainty associated with these two factors has not been directly investigated using satellite altimeter information over Antarctic sea ice. This work provides insights from a case study region, McMurdo Sound Antarctica. Snow on Arctic sea ice has been investigated in more detail and over a longer period than the Antarctic so climatologies can be produced (Warren et al., 1999). These datasets in combination with satellite altimetry, and suitable airborne investigations have permitted the completion of pan-Arctic thickness assessments (Kurtz et al., 2014, Laxon et al., 2013, Kwok and Cunningham, 2008). The research community lacks snow climatology information in the Southern Ocean, though dedicated basin-scale snow depth assessments are available via passive microwave sensors (Markus and Cavalieri, 2006). Continual improvements in our monitoring ability are key to support the current ESA satellite altimeter missions, CryoSat-2 (CS-2) and Sentinel-3 and NASA's laser altimeter mission ICESat-2. To date only AMSR-E passive microwave data have been used in combination with altimetry to estimate sea ice thickness. The AMSR-E algorithm's accuracy is decreased by rough sea ice and deep and complex snow (Kern and Ozsoy-Çiçek, 2016, Kern et al., 2011, Worby et al., 2008b, Stroeve et al., 2006), both typical characteristics of the Antarctic sea ice cover. Using laser altimetry, some investigators have assumed zero ice freeboard (Kurtz and Markus, 2012), that is, the snow loading forces the ice surface to the waterline, negating the need for snow depth data. Thickness estimates using this approach are likely biased low and although this simplification provides valuable insights, it does not provide sea ice thickness at the desired accuracy. This work is motivated by the necessity for a comprehensive understanding of the usefulness of snow products in the Southern Ocean, and the need to investigate new avenues for producing snow depth products over Antarctic sea ice. Here we make use of a detailed *in situ* dataset to assess modelling and satellite approaches to construct snow depth over the 2011 sea ice growth season. In a first attempt over Antarctic fast ice, using a high-resolution snow accumulation model called SnowModel (Liston and Elder, 2006a) and synthetic aperture radar imagery, we are able to establish when the sea ice fastens and accumulate snow from those dates for three areas of fast ice in McMurdo Sound in the south-western Ross Sea. The high-resolution model results are compared to snow products from two other independent datasets, the first ERA-Interim (ERA-I) precipitation and the second satellite passive microwave snow depth from AMSR-E. With these different snow depth datasets we infer sea ice thickness via freeboard measurements from CS-2. The interaction of radar energy with the snow pack is highly complex and here we take a simplified approach given the surface height has already been established by the ESA retracking procedure. Given the uncertainty of the position of the retracking point with reference to the height above sea level, we assume

different penetration depths into the snowpack by varying the proportion of ice and snow that
represents freeboard. We compare the inferred CS-2 thicknesses with *in situ* information.

## 2 Study area, field and satellite data

### 2.1 McMurdo Sound and field data

A detailed *in situ* sea ice measurement campaign was carried out in November 2011 on the fast
ice in McMurdo Sound (Fig. 1). This involved sea ice thickness, freeboard and snow
depth/snow density measurements at 39 sites. Freeboard was measured 5 times in a cross
profile at each site, once at the centre of the cross and once at the terminus of each line, as was
thickness. Mean snow depths for each *in situ* site represent 60 individual snow depth
measurements over that same cross-profile at 50 cm intervals. Snow density was measured at
18 sites, well distributed across the area, the mean of these sites is used for this analysis unless
stated otherwise. A full overview of the measurement procedure is provided in Price et al.
(2014). Additional *in situ* measurements of sea ice thickness are included in the analysis, two
measurements taken at one location in McMurdo Sound in July and November. Assuming a
constant growth rate between these measurements they are used in section 5 as a comparison
to CS-2 inferred sea ice growth rates. More detail on how the *in situ* thickness measurements
are used and how they should be interpreted is provided in section 5.

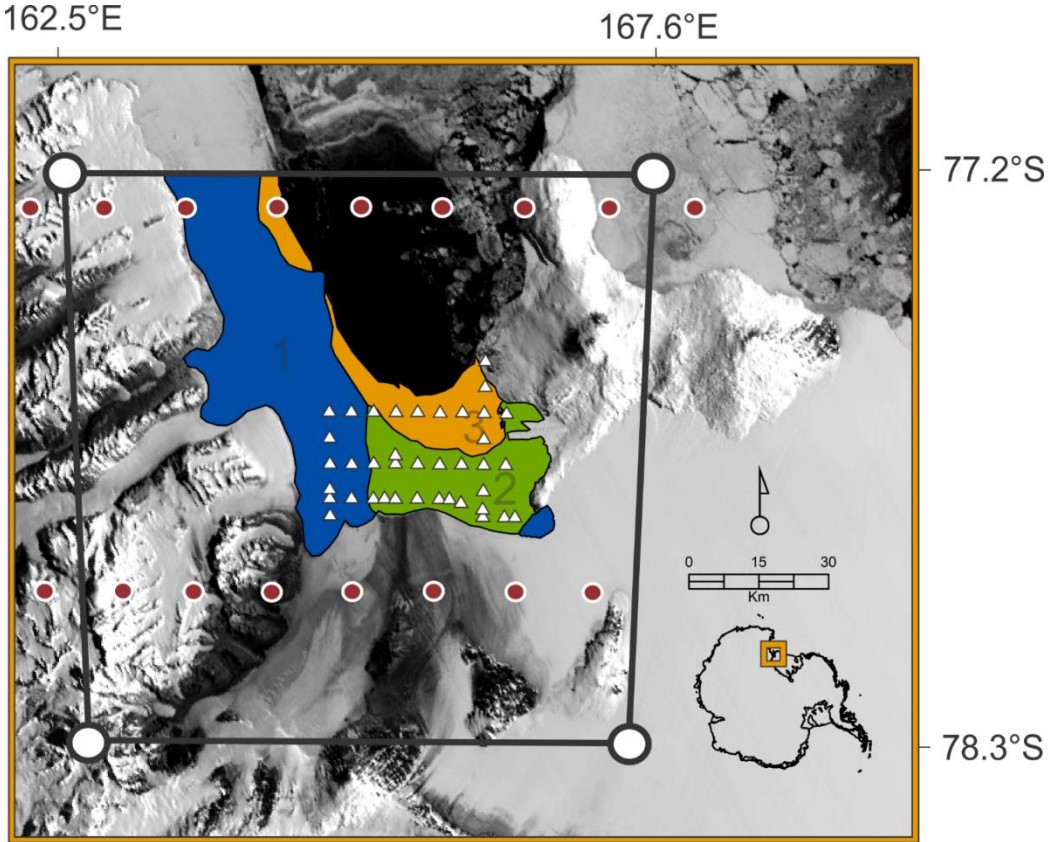


**Figure 1.** McMurdo Sound study area with each fastening area as identified by Envisat radar imagery:
area 1 – 01/04/2011 (Blue), area 2 – 29/04/2011 (Green), area 3 – 01/06/2011 (Orange) and SnowModel
domain bounded by the black box. Fastening areas are superimposed on a MODIS image acquired on
15 November at the time of maximum fast ice extent in 2011. The locations of 39 measurement sites
used to produce the *in situ* snow and sea ice statistics are shown as white triangles. The centre points of
each ERA-I 0.75° x 0.75° grid cell in the vicinity of the study area are displayed as red circles.

## 2.2 Envisat

The sea ice freeze-up provides a point from which snow can begin to accumulate on the sea ice surface. Freeze-up could be identified using passive microwave information, but this data does not provide the spatial resolution to segment the sea ice area appropriately for SnowModel's 200 m resolution. In McMurdo Sound during the freeze-up period, pack ice is generally advected north out of the study area unless it fastens. In addition to floe movement, before fastening occurs, snowfall is subject to uncertainty from flooding events and snow loss to leads, influences on the eventual snow depth that we have no way of accurately monitoring. With the resolution restriction in mind and these uncertainties, we have selected the sea ice fastening date to begin snow accumulation. To identify the dates and the pattern in which the sea ice fastens across the study area, we use a string of *C*-band Advanced Synthetic Aperture Radar (ASAR) images from Envisat acquired in Wide Swath mode. We process these files using GAMMA Software to produce ASAR imagery with a spatial resolution of 150 x 150 m. By comparing motion and patterns between sequential images we are able to identify three areas that fastened independently of one another. The first area of fast ice was established by 1 April (area 1 – Fig. 1), by the end of April, a second area of fast ice had formed along the southern extremity of the Sound (area 2 – Fig. 1), and by the beginning of June, a third area had fastened (area 3 – Fig. 1). The largest gap in the Envisat image string is 8 days but no large gaps are found around key fastening dates. The typical spacing is 1-2 days so we have confidence we have reduced our error in the fastening date to less than 2 days. These three areas persisted for the winter and when combined, made up the fast ice area present in late November when *in situ* measurements were made.

## 2.3 AMSR-E

The EOS Aqua Advanced Microwave Scanning Radiometer (AMSR-E) was operational from December 2002 until 4 October 2011. The snow depth product provided by NSIDC (https://nsidc.org/data/AE_SI12/versions/3#) is provided at a 12.5 x 12.5 km$^2$ polar stereographic projection and reported as a 5-day running mean, that mean inclusive of that day and the prior 4 days. We remove data where ice concentrations are lower than 20%. Gridded snow depth values are calculated using the spectral gradient ratio of the 18.7 and 36.5 GHz vertical polarisation channels. For snow free sea ice the emissivity is similar for both frequencies. Snow depth increases attenuation from scattering but is more pronounced at 36.5 GHz than at 18.7 GHz, resulting in higher brightness temperatures at 18.7 GHz (Comiso et al., 2003, Markus and Cavalieri, 1998). Using coefficients derived from a linear regression of *in situ* snow depth measurements on microwave data, and a 36.5-18.7 GHz ratio corrected for sea ice concentration, snow depth can be estimated (Comiso et al., 2003). Snow depth retrievals are restricted to dry snow only and to a depth of less than 50 cm. Variable snow properties including snow grain size, snow density and liquid water content influence microwave emissivity from the sea ice surface and the algorithm is reported to have a precision of 5 cm (Comiso et al., 2003) . Given the extreme southern latitude of the study area, snow conditions throughout this study were very dry, supported by snow pit analysis on the sea ice in November with no wet snow or lensing observed. AMSR-E cells are included in the analysis if over 50% of the cell lies within the fast ice mask, and segmented into each fastening area by that same criteria. 22 AMSR-E cells are used and due to the instrument failure in early October 2011, data for the last two months of this investigation are unavailable.

### 2.4 CryoSat-2

CS-2 was launched in 2010 and houses a *Ku*-band radar altimeter (centre frequency 13.6 GHz). The altimeter has an approximate footprint size of 380 m x 1560 m and samples along-track at 300 m intervals. The instrument has three modes and over the coastal Antarctic operates its interferometric (SIN) mode. This mode uses both of the satellite's antennas to identify the location of off-nadir returns accurately. This is not the dedicated sea ice mode, but it is still suitable for sea ice freeboard retrieval (Price et al., 2015; Armitage and Davidson, 2014). In section 5, to assess the usefulness of the evaluated snow products, we infer sea ice thickness from CS-2 freeboard measurements.

The ESA L2 baseline C SIN mode (SIR_SIN_L2 – available at: http://science-pds.cryosat.esa.int/) data set provides a retracked height for the surface over sea ice and this initial measurement is termed radar freeboard. The processing closely follows that described in Price et al. (2015), but to reduce noise, two modifications are made to achieve more detailed scrutiny of the CS-2 height retrievals. The first is a more stringent exclusion of off-nadir elevation retrievals, the threshold is halved from ± 750 m to ± 375 m; data located at greater distances from nadir are discarded. The second is the rejection of freeboard measurements of less than -0.24 m and greater than 0.74 m. Following Schwegmann et al (2016) the ± 0.24 m accounts for speckle range noise in the CS-2 data and the + 0.5 m threshold additionally incorporates an expected maximum sea ice freeboard of 0.5 m for fast ice in McMurdo Sound (as measured *in situ* in 2011). Each CS-2 radar freeboard measurement is cross-referenced to fastening areas 1, 2 and 3 and assigned a snow depth (*Ts*) value from the described snow products. From the ESA retracked product there is currently no consensus on what surface the radar freeboard represents over sea ice, the air-snow interface, the snow-ice interface or an undefined interface between the two. Laboratory experiments (Beaven et al., 1995) and comparisons of other radar altimeter systems with *in situ* measurements (Laxon et al., 2003) suggest the snow-ice interface is detected. It is clear that the presence of snow influences the CS-2 height retrieval, but precisely how, is dependent on the surface roughness (Kurtz et al., 2014; Hendricks et al., 2010; Drinkwater, 1991), its depth (Kwok, 2014) and its dielectric properties (Hallikainen et al., 1986). The mean depth of the dominant backscattering surface measured using a surface based *Ku*-band radar over snow covered Antarctic sea ice was around 50% of the mean measured snow depth, and the snow-ice interface only dominated when morphological features or flooding were absent (Willatt et al., 2010). Wingham et al. (2006) indicate the snow-ice interface is represented by the ESA retracked height. No other information is available about the assumptions made here, only that for diffuse echoes in SAR processing, for baseline C, a new retracker was implemented (Bouffard, 2015). It is unclear what the original retracking assumptions are for any retrieval mode and if any changes were made to SIN mode for baseline C. A prior study of CS-2 waveform behaviour over the same study area found ESA L2 freeboard to be located between the air-snow and snow-ice interface (Price et al., 2015). Given this uncertainty we apply a simple methodology to discover the range of thicknesses as inferred via this CS-2 data. We explore this possible range by changing the amount of snow and ice assumed to represent the freeboard measurement in the thickness equation. There is no physical change to the actual radar penetration, the inferred thickness is simply altered mathematically using a varying penetration depth (*Pd*) into the snow pack. Equation 1 assumes that the snow surface is detected, equation 2 that the sea ice surface is detected and equation 3 that an arbitrary surface at varying *Pd* values into the snow pack (0.02

m, 0.05 m, 0.10 m, 0.15 m, 0.30 m and 0.50 m - or to the snow-ice interface, whichever criteria is met first) represents the retracking point. The radar freeboard is corrected when snow is present and penetration is assumed (i.e. $Pd > 0$) for the reduction of the speed of the radar wave through the snow pack following the procedure described in Kurtz et al (2014). We derive sea ice thickness ($Ti$) using the newly corrected freeboard ($Fb$) and the described equations;

$$T_i = \frac{\rho_w}{\rho_w - \rho_i} Fb \; - \; \frac{\rho_w - \rho_s}{\rho_w - \rho_i} T_s \tag{1}$$

$$T_i = \frac{\rho_w}{\rho_w - \rho_i} Fb \; + \; \frac{\rho_s}{\rho_w - \rho_i} T_s \tag{2}$$

$$T_i = \frac{\rho_w}{\rho_w - \rho_i} Fb \; - \; \frac{\rho_w - \rho_s}{\rho_w - \rho_i} T_s \; + \; \frac{\rho_w}{\rho_w - \rho_i} Pd \tag{3}$$

where $\rho_w$ (1027 kgm$^{-3}$), $\rho_i$ (925 kgm$^{-3}$) and $\rho_s$ (385 kgm$^{-3}$) are the densities of water, sea ice and snow respectively. $\rho_w$ is informed by an unpublished time series of surface salinity measurements taken from October 2008 to October 2009 along the front of the McMurdo Ice Shelf. The range in $p_w$ during this period is less than 1 kgm$^{-3}$. The $\rho_i$ value used here is in the middle of the measured range in McMurdo Sound, the use of which is discussed in Price et al. (2014). $\rho_s$ is the mean value taken from 18 of the 39 *in situ* sites where snow density was measured.

**3 Atmospheric models for snow accumulation**

**3.1 High resolution model**

SnowModel is a numerical modelling system with four main components: (1) MicroMet, a quasi-physically-based, high-resolution meteorological distribution model (Liston and Elder, 2006b) (2) Enbal, a surface energy balance and snowmelt model (Liston et al., 1999) (3) SnowTran-3D, a wind driven snow redistribution routine (Liston et al., 2007, Liston and Sturm, 1998) and (4) SnowPack, a multilayer snow depth and water-equivalent model (Liston and Sturm, 1998). The main objective of MicroMet is to provide seamless atmospheric forcing data, both temporally and spatially to the other SnowModel components. MicroMet is capable of downscaling the fundamental atmospheric forcing such as air temperature, relative humidity, wind speed, wind direction, incoming solar radiation, incoming longwave radiation, surface pressure, and precipitation. Other SnowModel submodels simulate surface energy balance, and moisture exchanges including snow melt, snow redistribution and sublimation. SnowModel also incorporates multilayer heat and mass-transfer processes within the snow (e.g. snow density evolution).

SnowModel is capable of initializing with both *in situ* and gridded model data and has been evaluated in many geographical locations including Greenland and Antarctica (Liston and Hiemstra, 2011; Liston and Hiemstra, 2008; Liston and Winther, 2005; Mernild et al., 2006). To the authors knowledge, and at the time of writing this is only the second application of SnowModel in a sea ice environment. Liston et al. (2018) applied SnowModel with an

additional component that accounted for snowdrifts and snow dunes, at very high spatial resolution over Arctic sea ice with positive results.

SnowModel requires topography, land cover and various atmospheric forcing. The minimum meteorological requirements of the model are near-surface air temperature, precipitation, relative humidity, wind speed and direction data from Automatic Weather Stations (AWS) and/or gridded numerical models. Determining the influence of wind and other atmospheric forcing on snow distribution in a complex terrain requires the use of numerical atmospheric models. Many studies have demonstrated that high-resolution models are vital for simulating topographic and land-use impacts on wind, hydraulic jump and associated turbulence (Olafsson and Agustsson, 2009; Agustsson and Olafsson, 2007). For this research, hourly atmospheric forcing were generated by version 3.5 of the polar-optimized version of the Advanced Research Weather Research and Forecasting Model (WRF-ARW; Skamarock et al., 2008) known as Polar WRF (Bromwich et al., 2009) or PWRF (http://polarmet.osu.edu/PWRF) at 3 km horizontal resolution.

The WRF-ARW (hereafter, WRF) is a state-of-the-art model that is equipped with a fully compressible, Eulerian and nonhydrostatic dynamic core. This model uses Arakawa C-grid staggering in the horizontal and utilises a mass terrain-following coordinate vertically. Several physical parameterization schemes are available in WRF, and some of those used for this work are described below. The WRF single-moment 6-class microphysics scheme (WSM6; (Hong and Lim, 2006)) is a cloud microphysics scheme, which includes various water phases including graupel. This likely improves precipitation and cloud related predictions at higher spatial resolution. For radiation, the rapid radiative transfer model (RRTM;(Mlawer et al., 1997)) and the empirically based Dudhia short-wave radiation scheme (Dudhia, 1989) are used as the long and short wave radiation schemes, respectively. The Mellor–Yamada–Nakanishi–Niino (MYNN; Nakanishi and Niino, 2006, Nakanishi and Niino, 2004, Nakanishi, 2001) level-2.5 scheme is used to take into account subgrid-scale turbulent fluxes.

The Noah LSM (Chen and Dudhia, 2001) with four soil layers, which is able to handle sea-ice and polar conditions through modifications described below was chosen as the land surface model. Generally, mesoscale numerical models including WRF have simple representations for sea ice thickness and snow depth on sea ice. This shortcoming leads to an outstanding error in the simulation of the snow and mass balance in the polar regions. To address this issue, PWRF improved the representation of heat fluxes through snow and ice in the Noah LSM. Further, this version of PWRF modified sea ice and snow albedos and made it accessible to define spatially varying sea ice thickness and snow depth on sea ice [for further detailed information about PWRF see Hines et al. (2015)].

The models, PWRF and SnowModel are coupled in an off-line manner. This means that the PWRF model ran for the entire study period first, then SnowModel initiated based on the PWRF simulated atmospheric forcing and there is no feedback from SnowModel to the atmospheric model. In order to increase the spatial resolution of the PWRF outputs, before ingesting the atmospheric forcing to the SnowModel, PWRF gridded data are interpolated to a new grid, and then corrected physically according to topography using the MicroMet submodel. The spatial resolution of SnowModel is 200 m and its output is segmented into sea ice fastening areas as indicated by the Envisat imagery (Fig. 1). Model outputs are reported as hourly means beginning at 00:00 1st April 2011 and ending at 00:00 1st December 2011.

SnowModel outputs snow depth and swe. The model has a varying density over time. The swe
output is important as it allows comparison of the model to the other snow products which have
different density assumptions.
**3.2 Low resolution model**
ERA-I is a global atmospheric reanalysis product on a 0.75° x 0.75° grid available from 1
January 1989 (Dee et al., 2011). Precipitation data (mm water equivalent) are available at three
hourly intervals and are converted to snow depth when required using the average snow density
of 385 kgm$^{-3}$ measured *in situ* in 2011. Using splines we interpolate the coarse resolution ERA-
I grid and provide a 10 x 10 grid over the study area with a cell resolution of 12 km. The
reanalysis does not account for snow transport but with the interpolated grid we are able to
segment the model for sea ice fastening dates and begin snow accumulation at the correct time.
We average the three hourly outputs, the reported ERA-I data are daily averages for each
fastening area.
**4 Snow product evaluation**
When the three snow products are compared to one another, or to *in situ* measurements, all
snow depths are reduced to snow water equivalent (swe) via their respective densities to
remove any bias associated with varying density between snow datasets. SnowModel provides
a swe output via a time varying snow density during the model run, AMSR-E snow depths are
reduced to swe using average *in situ* measured snow density in November, and ERA-I
precipitation is provided as swe in its original format.  The SnowModel evaluation is split into
three parts, firstly, an accumulation time-series is presented for each snow product segmented
by each fastening area, and this time series is the mean snow depth for each product within
each area (Fig. 2). Secondly, selected SnowModel grid cells are directly compared to spatially
coincident *in situ* measurement sites in November (Fig. 3) and thirdly, the SnowModel and
ERA-I distributions are plotted as maps at the end of the model run for spatial comparison (Fig.
4). The model swe values used for direct comparison to *in situ* measurements in Figures 3 and
4 are the mean at each site between 25$^{th}$ November and 1$^{st}$ December, the period over which *in*
*situ* measurements were made.
The SnowModel mean swe for all areas at the end of the simulation is 2 cm higher than *in situ*
swe mean. However, SnowModel clearly presents two very different snow accumulation
patterns, one in the west covering area 1 and one in the east covering areas 2 and 3. Mean swe
values in area 1 reach a maximum of 2 cm during the 8-month study period while in areas 2
and 3 they are in excess of 10 cm. This broad spatial distribution produced by SnowModel
compares well with *in situ* measurements and general observations in November 2011, which
recorded an increasing gradient in snow depth from west to east (Fig. 4). However, when each
fastening area is directly compared to *in situ* means for those areas, swe is underestimated in
area 1 (2 cm < *in situ*), slightly overestimated in area 3 (1 cm > *in situ*) and substantially
overestimated in area 2 (5 cm > *in situ*) (Fig. 2). Only modelled swe in area 3 falls within the
standard deviation of the *in situ* mean. In the east, snow depth increases are noted in mid-May,
mid-June, early-July, early and mid-August and late-September. The snow depth evolution in
the west of the Sound over area 1 follows a separate pattern with negligible increases in mid/late
April, mid-May, mid-July, late-September and early-November. When coincident pixels are
directly compared to *in situ* data with coincident pixels SnowModel overestimates swe in the
study area and therefore the model has better agreement with *in situ* maximum values ($r^2 =$

0.56) than with the mean ($r^2 = 0.53$) or minimum ($r^2 = 0.30$) values (Fig. 3). It is important to note the importance of redistribution by wind which is provided by SnowModel. The consequences of neglecting this influence on snow accumulation in the study region are clearly demonstrated in Figure 4. Figure 4a displays the accumulated precipitation from MicroMet, while this is built on in Figure 4b with the inclusion of the other SnowModel components. Over eastern areas of the study region, the MicroMet precipitation output as a standalone product provides swe values double that of the highest swe measured *in situ*. Although vastly improved, the general overestimation of swe by SnowModel is clearly visible in Figure 4b. Values in the eastern most section of the sea ice cover in McMurdo Sound, adjacent to Ross Island are in the order of 20 to 35 cm swe. These values are all larger than the highest *in situ* measured swe of 17.7 cm and for large areas, they still remain over double the measured value. In the central area of the Sound, modelled swe decreases in agreement with measured swe with 5 *in situ* sites agreeing within ± 0.5 cm of SnowModel swe (Fig. 3 and Fig. 4b). The western region of sea ice in fastening area 1 has far less measured snow. The model produces this well but values are too low. The extremes, where there is a lot of snow and where there is very little snow both seem to be exaggerated by the model.

Unlike SnowModel or the *in situ* distribution in late November AMSR-E swe follows a similar pattern over time in all fastening areas. For areas 2 and 3, May through June, AMSR-E and SnowModel produce similar swe values, agreeing within 1.5 cm in areas 2 and 3. In area 1 AMSR-E swe fluctuates but is typically about 2.5-3 cm higher than SnowModel. As the growth season progresses AMSR-E remains significantly lower than SnowModel swe in areas 2 and 3, by up to 10 cm. swe values are higher in area 2 than area 3 in agreement with SnowModel. However, in area 1 swe values are four times larger than SnowModel. Most importantly, the longitudinal swe gradient indicated by SnowModel and supported by *in situ* data is opposite when measured using AMSR-E (i.e. swe is higher in the west than in the east for the duration of the times series). As the AMSR-E instrument failed in early October, we are unable to validate it with *in situ* measurements. ERA-I also produces a different snow distribution to SnowModel and *in situ* data (Fig. 4c) with an area of lower swe values in the central area of the fast ice and higher swe values over the western and eastern areas. The mean deviation over the entire study area from *in situ* measurements is 20 cm swe. ERA-I swe values are over double that of SnowModel for areas 2 and 3 and an order of magnitude higher for area 1 (Fig. 2). The ERA-I temporal snowfall pattern is the same between all areas and is similar to that produced by Snow Model in areas 2 and 3.

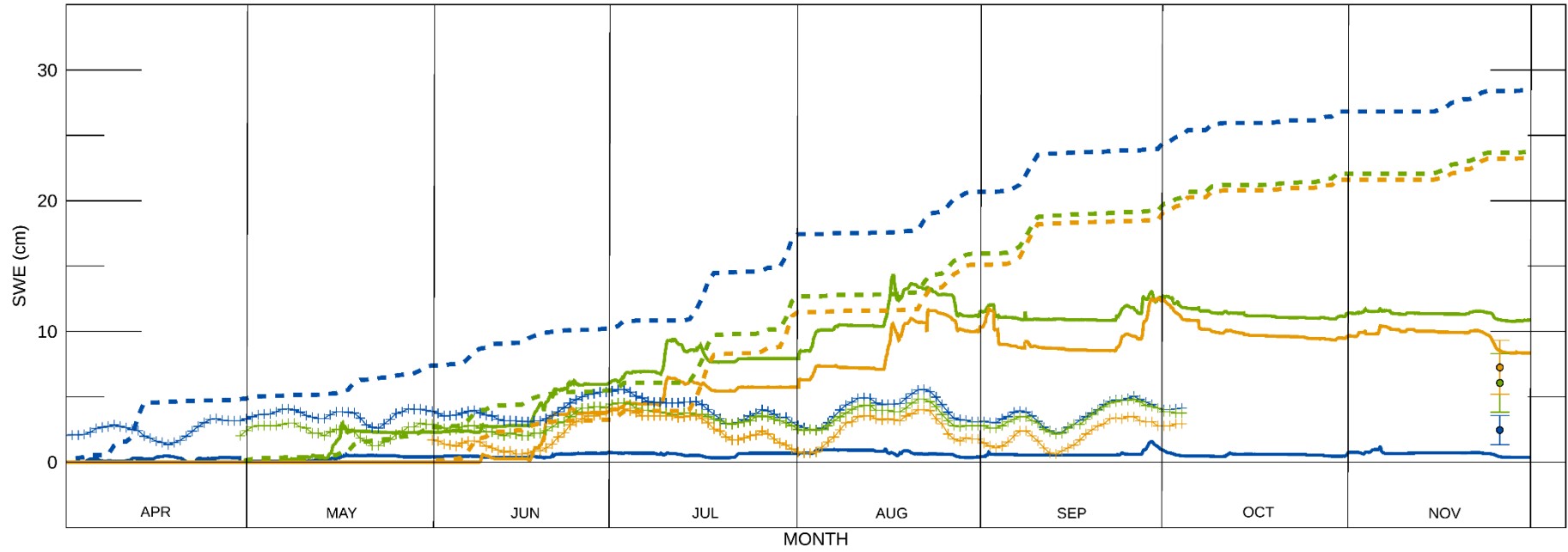

**Figure 2.** SnowModel hourly (solid lines), ERA-I daily (hashed lines) snow water equivalent (swe) accumulation and AMSR-E daily snow depth (crosses) converted to swe for fastening areas 1 (blue), 2 (green) and 3 (orange). The mean *in situ* swe and standard deviations for each area are displayed as circles at the end of November and colour coded to their respective fastening areas.

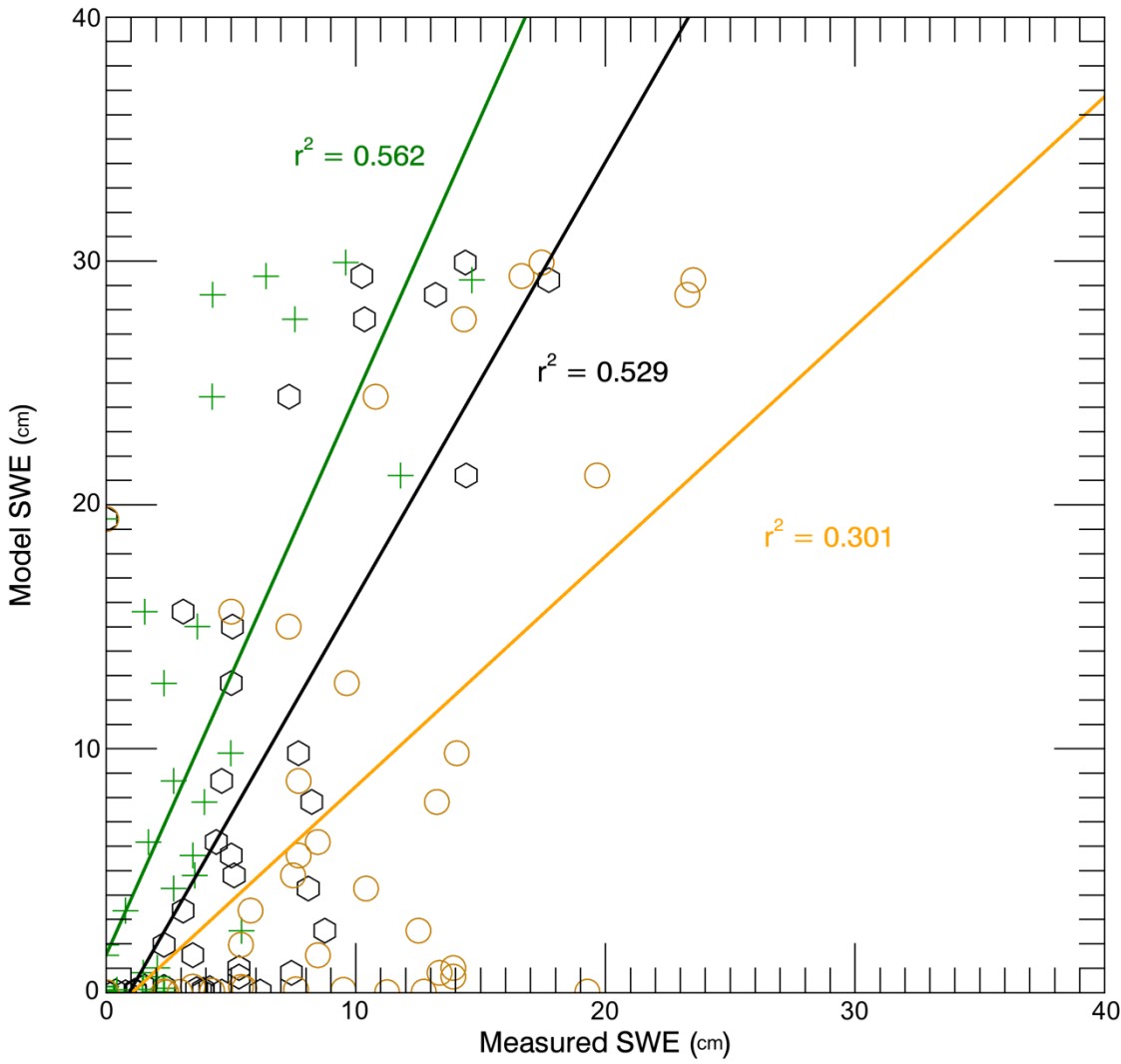

**Figure 3.** Mean (black), maximum (green) and minimum (orange) *in situ* measured snow water equivalent (swe) for each site against mean SnowModel swe at each coincident model cell for the *in situ* measurement period.

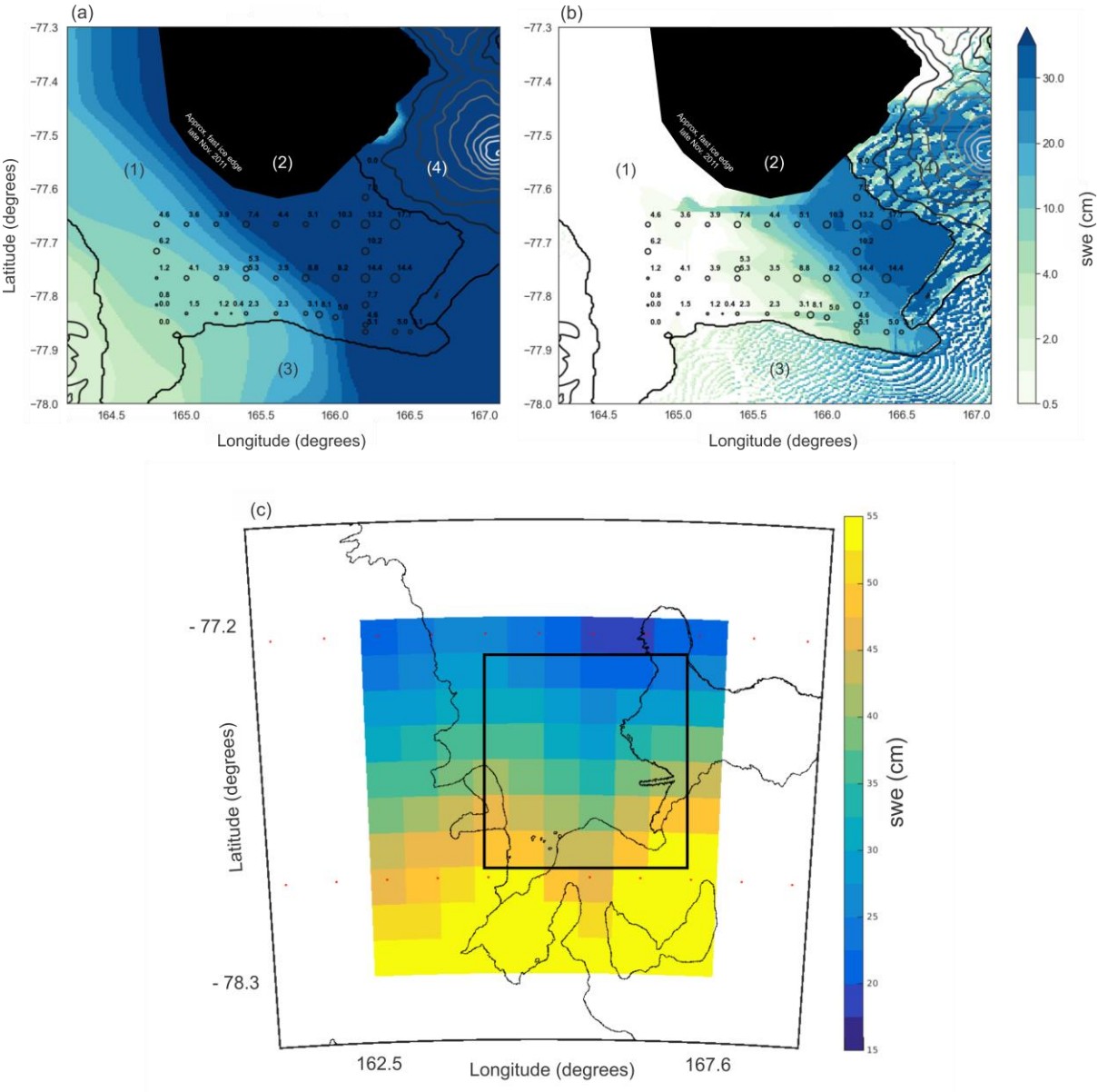

392

**Figure 4.** (a) MicroMet swe distribution and (b) SnowModel swe distribution in McMurdo Sound, with (1) fast ice, (2) open water/pack ice, (3) McMurdo Ice Shelf, (4) Ross Island identified. The model swe distribution is the mean of the simulation over the *in situ* measurement period (25th November-1st December). The *in situ* measurements were converted to swe via the density measured at each site, if no measurement was taken (21 sites) the average *in situ* snow density was used (385 kgm$^{-3}$). *In situ* measurement locations are shown as black circles and are the mean of the 60 snow measurements taken at each site. The circle sizes are weighted for swe to allow visualisation of the decreasing swe distribution from east to west. Elevation contours are spaced at 400 m intervals; Mt Erebus (3,794 m) is the dominant topographic feature on Ross Island to the east of the fast ice. (c) The interpolated 10 x 10 ERA-I grid with 1$^{st}$ December accumulation total, the boundary of the SnowModel inset from (a) is shown as the black box. The ERA-I centre points of the original grid are displayed as red dots.

404

405

406

## 5 Sea ice thickness

In this section, we review the usefulness of the snow products by using them as inputs to equations 1-3 and infer sea ice thickness in McMurdo Sound through the growth season. Snow information, coincident in space and time for each CS-2 measurement is retrieved from the SnowModel and AMSR-E products as snow depth, while ERA-I swe is converted to snow depth using the mean *in situ* measured density.

Sea ice thickness inferred from altimetry in McMurdo Sound will be influenced by the buoyant sub-ice platelet layer (Price et al., 2014). The *Fb* measurement used to infer thickness is representative of the solid sea ice and the layer of sub-ice platelets attached below. Therefore, comparisons to *in situ* thickness referenced in this work actually refer to the 'mass-equivalent thickness', that is, the resultant thickness taking account of both the solid sea ice and the sub-ice platelet layer (sub-ice platelet layer multiplied by the solid fraction). The only exception to this is the red line in Fig. 5 which is a linear fit between two measurements of consolidated sea ice thickness in July and November 2011 used here to show the sea ice thickness growth rate for comparison to CS-2 thickness trends.

From equations 1-3, sea ice thickness is highly sensitive to the snow-ice ratio for the measured freeboard. This results in a large range in sea ice thickness for all snow products through the growth season (Fig. 5). This range in inferred thickness is driven by the amount of snow produced by the models as Eq. 1 and Eq. 2 subtract and add the product of this value in their second terms respectively. As the snow depth increases, in some cases to higher values than the measured freeboard the *Pd* simply provides a correcting factor for this discrepancy. The AMSR-E derived thickness trend is not comparable to the model output trends as the last two months are missing. However, it is useful to highlight the importance of the snow-ice freeboard ratio. AMSR-E snow depths remain relatively stable for the duration of the study. Because of this, the ratio of ice to snow above the waterline remains very similar. In the case of the models, snow depths gradually increase and snow makes up an ever increasing proportion of mass above the waterline. If the air-snow interface (Eq. 1) is taken to represent *Fb* then the trend in sea ice thickness through the growth season is negative for SnowModel and ERA-I derived thicknesses and if the snow-ice interface (Eq. 2) is assumed the trend is too positive. The trends are more extreme for the ERA-I estimates simply because the snow loading is greater. The ranges in sea ice thickness estimated with SnowModel as the snow depth input are substantially smaller than ERA-I (Fig. 5), but still have a larger range than the mean discrepancy from *in situ* measurements might suggest (Fig. 2). This is driven by CS-2 retrievals over the eastern areas of fastening areas 2 and 3 where swe values are high, especially towards the end of the growth season (Fig. 4b). The range in uncertainty between Eq. 1 and Eq. 2 derived thickness as means of available data for the entire growth season are 1.08 m, 4.94 m and 1.03 m for SnowModel, ERA-I and AMSR-E respectively. The mean CS-2 derived thickness values for November using Eq.1 and Eq. 2 are 1.02 m (-2.98 m) for SnowModel (ERA-I) and 2.62 m (6.59 m) for SnowModel (ERA-I) respectively compared to an *in situ* thickness of 2.4 m. The trends that result in a November thickness supported by the *in situ* measurements are those that assume penetration into the snow cover, analogous with the retracked surface representing a surface between the air-snow and snow ice interfaces. For thicknesses derived using the models to match *in situ* thickness large *Pd* values of 0.5 m are required given the higher snow depth values. These values are lower for AMSR-E as the snow loading is less.

The differences in the snow depths from each model result make it difficult to constrain what
*Pd* value provides CS-2 thicknesses that agree best with measured thickness. To assess the
penetration uncertainty further we use interpolated *in situ* measurements for snow depth as
input to the sea ice thickness calculation. We reduce the CS-2 measurements used in this
comparison to the same area bounded by *in situ* measurements. The total range in estimated
sea ice thickness using interpolated *in situ* snow depth between equations 1 and 2 is 1.7 m. For
*Pd* values 0.02 m through 0.20 m the best agreement between *in situ* thickness and CS-2 derived
thickness is found between 0.05 and 0.10 m (Fig. 6 – third column, 'In situ'). The CS-2
thickness is only 0.02 m thicker than *in situ* thickness for this particular dataset when *Pd* = 0.07
m. The range in SnowModel derived thickness between Eqs. 1 and 2 is nearly 4 m while the
range when using the ERA-I data set is very large at 5.7 m (Fig. 6). Again this large range in
thickness reflects the higher average snow depth produced by ERA-I. The deeper snow creates
a larger range of snow-to-ice ratios for freeboard.

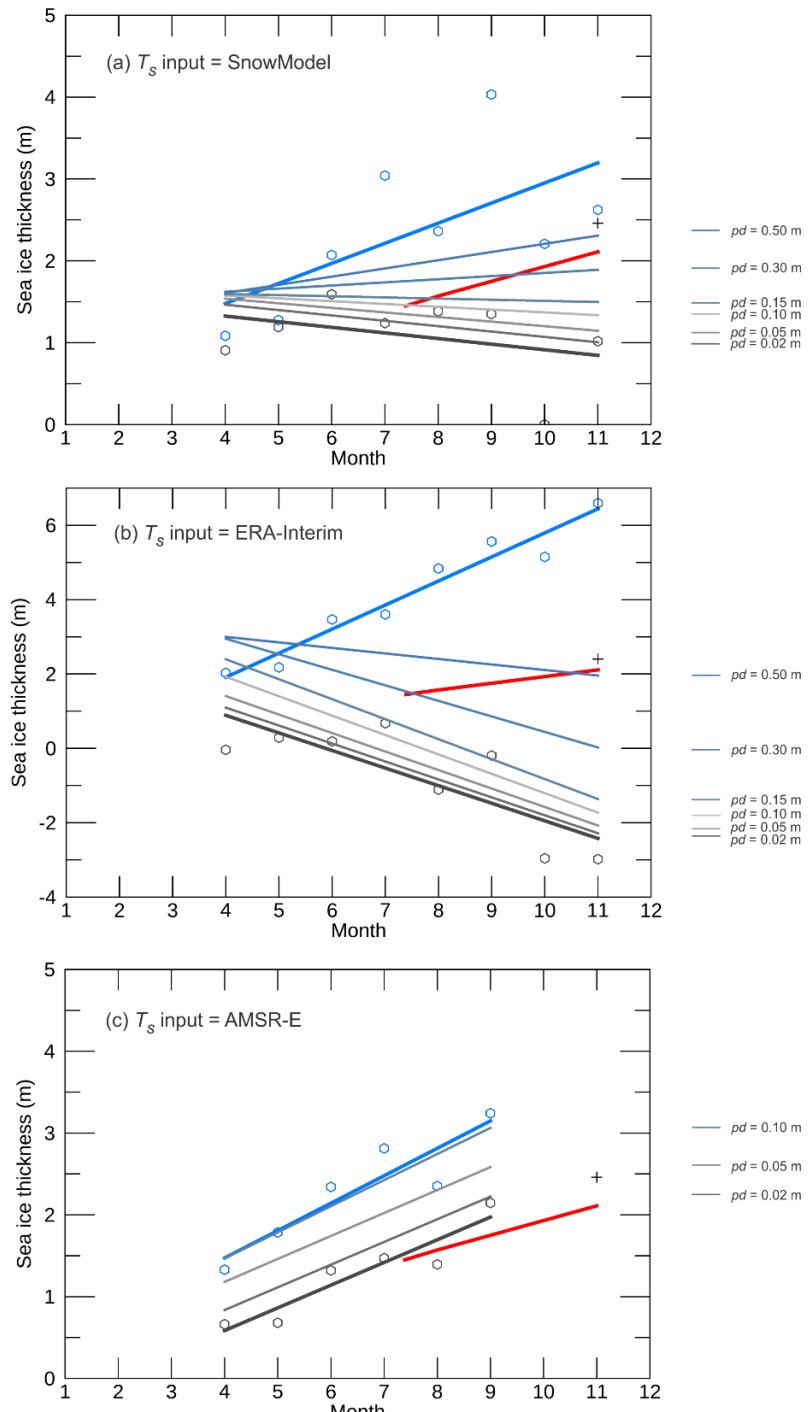


**Figure 5.** Sea ice thickness trends derived by CS-2 freeboard measurements with snow data provided
by (a) SnowModel, (b) ERA-I and (c) AMSR-E. Grey dots and bold linear fit are sea ice thickness
calculated using equation 1, blue dots and bold linear fit using equation 2 and thin lines between them
equation 3 with varying penetration factors (*Pd*). The red line shows sea ice thickness from *in situ*
measurements of consolidated sea ice thickness with a tape measure taken in July and November in one
location in the south of McMurdo Sound joined assuming a constant growth rate. The black plus sign
is the mean 'mass-equivalent thickness' from all *in situ* measurements in November. This is slightly
thicker than the end of season thickness indicated by the red line given it takes account of the influence
of the sub-ice platelet layer. This black plus sign is what CS-2 thickness should be compared to (see
text).

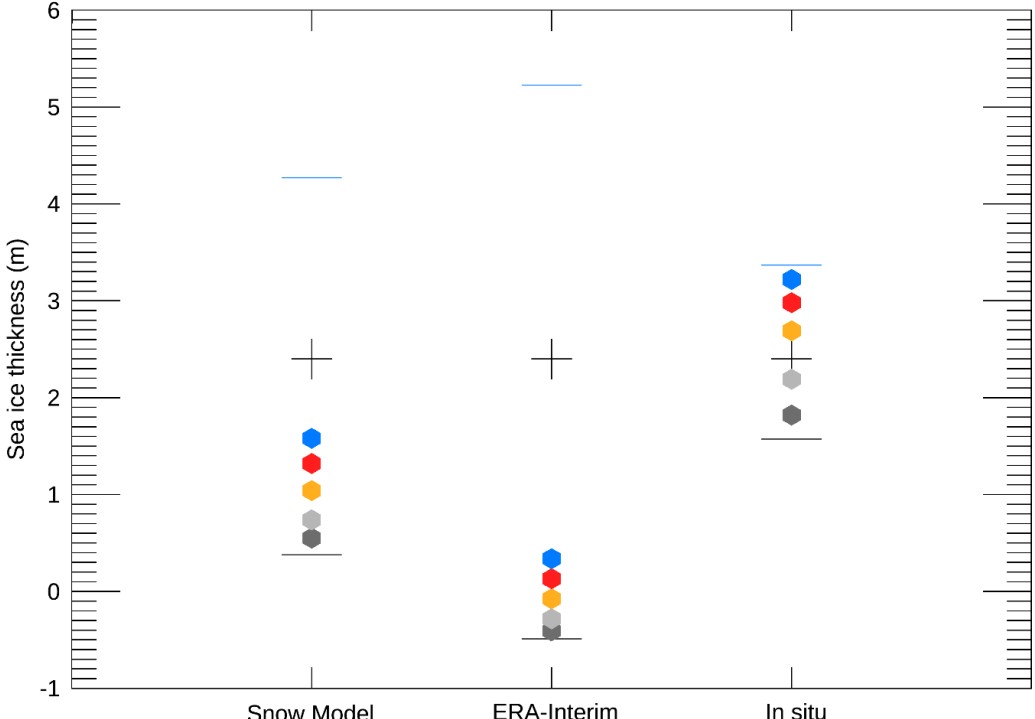

476

**Figure 6.** The range in CS-2 derived sea ice thickness in November using snow inputs from SnowModel and ERA-I compared to snow input from *in situ* interpolated snow depths. Thickness derived from equations 1 and 2 are shown with the grey and blue lines respectively and for equation 3 the dots are colour coded for different penetration depths (*Pd*); dark grey = 0.02 m, light grey = 0.05 m, orange = 0.10 m, red = 0.15 m and blue = 0.20 m. Black plus signs show *in situ* 'mass-equivalent thickness'. This comparison is produced from all CS-2 data height retrievals available over the *in situ* measurement area in November (*n* = 279).

## 6 Discussion

In this section, the performance of the snow depth retrieval methods and CS-2 thickness uncertainty is evaluated. We briefly discuss their future applicability to larger Antarctic sea ice areas.

Any method attempting to accumulate snow on sea ice requires the establishment of a starting date from which a sea ice surface is present. This approach used Envisat ASAR imagery and motion between scenes to identify when the sea ice fastened. Freezing may have started prior to the fastening date but the authors are unaware of any other method to monitor sea ice formation at the required spatial resolution for SnowModel. Sea ice could have begun to form slightly before this date, which, assuming a net gain in snow would result in an improvement in SnowModel's performance in area 1, but increased separation between *in situ* validation and SnowModel in areas 2 and 3. ERA-I performance would be worse in all cases, AMSR-E would not be impacted as it is a real-time snow depth measurement. In larger open water areas, passive microwave sea ice concentration information could be used to establish the formation date. Detail would be lost via this method given the high (200 m) resolution of SnowModel against the coarser resolution passive microwave data. Early snow fall on more dynamic pack ice will also be subject to flooding, sea spray (both likely to result in snow-ice formation) and loss to leads. These uncertainties must all be considered in future work.

Modelled snow depths have been evaluated in previous work over Antarctic sea ice (Maksym and Markus, 2008), but the study produced precipitation data while this assessment takes the next step by using a model that accounts for surface transportation, a significant redistribution mechanism in the Antarctic. Without this model component included the precipitation provided by MicroMet (downscaled PWRF) provides very poor estimates of snow depth on sea. Leonard and Maksym (2011) report that over half of precipitation over the Southern Ocean could be lost to leads and the application of any model to construct snow depth on sea ice in open sea areas will need to account for this. In coastal regions, local topography will also play a key role, such is the case in McMurdo Sound where Ross Island acts to encourage snow accumulation on the eastern portion of the sea ice cover. This was well replicated in SnowModel although the overestimation of snow was driven by unrealistic values in this area, the model likely accumulating too much snow due to this topographic barrier. Smaller scale snow features such as snow drifts and snow dunes should also be accounted for in future work, as applied in a recent study by Liston et al. (2018). These meter-scale features will be important to capture, especially to support compatibility with smaller satellite altimeter footprints, in particularly ICESat-2 (Markus et al., 2017). This work used fast ice to reduce the uncertainty associated with pack ice and used available *in situ* data to validate the snow products. To build on this approach, and make its application valuable in the Southern Ocean, sea ice motion within the SnowModel domain must be incorporated.

We find the ERA-I mean swe to be 20 cm higher than mean *in situ* swe in McMurdo Sound. In area 1 ERA-I swe is an order of magnitude higher than *in situ* swe, while in areas 2 and 3 it is over double the value. These create very high, unrealistic snow depths which causes a large range in CS-2 derived thickness using Eqs. 1-3. This is a very poor result and the product is inadequate to infer sea ice thickness when combined with altimetry data. Of further interest is that the clear longitudinal gradient in snow depth as indicated by SnowModel and measured *in situ* (November only) is not produced by ERA-I, swe values are lower in the central fast ice area and higher in the western and eastern areas. The performance of ECMWF reanalysis products over the satellite period has been reported as good when compared to Antarctic coastal stations (Bromwich and Fogt, 2004), but there is limited data available to assess the accuracy of these data over Antarctic sea ice. ERA-I ranked best among five assessed models for its depiction of interannual variability and overall change in precipitation, evaporation and total precipitable water over the Southern Ocean (Nicolas and Bromwich, 2011). Maksym & Markus (2008) used ERA-40 reanalysis for a snow assessment of the Antarctic sea ice pack but had difficulties in evaluating its accuracy. A first step to improve reanalysis results will be to incorporate snow redistribution (including snow loss to leads) and parameterisations for this could be built from wind vectors provided by the same reanalysis data.

In general, when compared to SnowModel, AMSR-E underestimates snow depth in areas 2 and 3 (eastern Sound) and overestimates snow depth in area 1 (western Sound). The snow distribution gradient from east to west is reversed in the AMSR-E dataset. Worby et al. (2008b) report that AMSR-E snow depths were significantly lower than *in situ* measurements on sea ice in the East Antarctic and that sea ice roughness is a major source of error using passive microwave retrieval techniques. However, they also conclude that when compared to basin-wide observations from ASPECT large differences of up to + 20 cm in the Weddell Sea and + 5-10 cm in the Ross Sea were noted in the AMSR-E snow depths. Vessels are restricted in their ability to sample in heavily deformed and thicker sea ice areas where the snow is typically

higher. Because of this, it is postulated that shipborne observations of *in situ* snow thickness were biased low in comparison to AMSR-E snow depth. More work is required to validate passive microwave snow depth estimates over Antarctic sea ice. No detailed sea ice surface condition survey was completed for this investigation, however from visual observations sea ice had clearly been subjected to dynamics in the west, whereas ice was very level in the east. It is possible that snow depth was underrepresented here by *in situ* measurements and that rougher sea ice in the west affected the AMSR-E retrieval algorithm. Because of the failure of the instrument, we are unable to compare AMSR-E snow depth directly to *in situ* measurements.

CS-2 has difficulty estimating freeboard over thin ice areas (Price et al., 2015, Ricker et al., 2014, Wingham et al., 2006). Here, at the beginning of the growth season CS-2 generally overestimates sea ice thickness with mean April values inferred using snow data from SnowModel and ERA-I of around 1 m (with the exception of AMSR-E assuming the air-snow interface is measured $T_i = 0.66$ m). Other investigations indicate that sea ice thickness in McMurdo Sound in April is between 0.5-0.8 m (Frazer et al., 2018, Gough et al., 2012, Purdie et al., 2006) . This represents a large obstacle to overcome for the application of CS-2 in the Southern Ocean as the mean thickness of Antarctic sea ice is only 0.87 m as reported from ship-based observations (Worby et al., 2008a). This supports the need for multisensor analysis, perhaps using methods already employed in the Arctic (Ricker et al., 2017, Kaleschke et al., 2012, Kwok et al., 1995). As discussed in section 2.4 assumptions must be made about what surface the freeboard measurement represents. In general, using the two modelled snow products (because trends from AMSR-E are incomplete), the thicknesses derived assuming the air-snow interface is freeboard are too thin and those assuming the snow-ice interface is freeboard are too thick, a simple consequence of the density dependent hydrostatic equilibrium assumption. By using the interpolated *in situ* measured snow depth as the snow thickness input to the thickness calculation, the error is minimised. With this, we find CS-2 thickness to correlate best with *in situ* thickness if *Pd* values are between 0.05-0.10 m. This is supported by other work in the study area (Price et al., 2015) who estimated the ESA elevation to be between the air-snow and snow-ice interfaces when sea surface height error was ruled out via a manual sea surface classification. Also recent work in the Arctic suggests that the height that represents radar freeboard provided by the ESA Level 2 product is closer to the air-snow interface than the snow-ice interface (King et al., 2018).

Having confidence in the results assumes that the sea surface height has been accurately identified for each CS-2 track. Freeboard errors from automated sea surface height identification were in the order of 0.05 m when compared to supervised procedures in the study area (Price et al., 2015). To eliminate this uncertainty throughout the study period the sea surface would need to be manually identified for each individual CS-2 track. This is not practical for basin-scale assessments and confidence needs to be built in the sea surface height identification algorithm. The modification of the sea surface height will apply a systematic increase or decrease in freeboard making each thickness from each assumption thicker or thinner. The freeboard measurements exhibit an unexpected decrease in October and November and it is impossible to discern whether this is forced by a sea surface height that is too high, or a change in the sea ice surface conditions that causes a decrease in the freeboard measurement, an additional uncertainty. More detailed *in situ* investigations, with surface roughness and snow characteristic statistics at the scale of the altimeter footprint are required

before a seasonally varying *Pd* can be applied with any confidence. As this analysis was
focused on the combination of independent snow products and CS-2 altimeter data, the range
in sea ice density has not been taken into account. We have confidence in the middle ground $\rho_i$
value used from previous work in McMurdo Sound (Price et al., 2014) but this is another source
of uncertainty for regional and basin-scale assessments.
**7 Conclusions**
This work has evaluated the ability of three independent techniques to provide snow depth on
fast ice in the coastal Antarctic. SnowModel accurately captures the *in situ* measured snow
distribution in November 2011 and produces a swe mean value that is 0.02 m above the mean
of *in situ* validation, but when sea ice is segmented by fastening date large deviations of up to
5 cm are present in the east where the model has overestimated snow depth. This accurately
captures the mechanism of snowfall and transport driven by the topography of Ross Island, but
the rates are higher than in reality. ERA-I swe is 20 cm higher than *in situ* measurements and
the gradient of the snow distribution produced by the analysis does not match that measured *in
situ*. A positive bias in accumulation should be expected from ERA-I as no snow redistribution
mechanism is included. Any future work making use of precipitation reanalysis over Antarctic
sea ice must include snow redistribution by wind, shown here by SnowModel to dramatically
improve results. AMSR-E snow depth information suffers from problems already documented
in the literature, and we find that its performance may have again been influenced by rough sea
ice. The snow distribution produced by AMSR-E was opposite to that provided by SnowModel
and measured *in situ* at the end of the growth season. We were unable to validate the instrument
due to its failure two months before the *in situ* data was collected. The uncertainty in the snow
depth estimates manifest themselves in the sea ice thickness estimates from CS-2. The range
in sea ice thickness uncertainty from the assumption that the snow surface or ice surface
represents freeboard, as means of the entire growth season are 1.08 m, 4.94 m and 1.03 m for
SnowModel, ERA-Interim and AMSR-E respectively. Using interpolated *in situ* snow
information, we find CS-2 freeboard measurements provided by the ESA retracker agree best
with *in situ* measured thickness if a dominant scattering horizon 0.07 m beneath the air-snow
interface is assumed, in agreement with recent literature. It is impossible to confidently
constrain this number without reducing uncertainty in the established sea surface height from
which the freeboard is estimated. This work demonstrates the need to reduce the uncertainty
associated with the ambiguity of the altimeter radar freeboard measurement over Antarctic sea
ice. Sea ice in McMurdo Sound is atypical of Antarctic pack ice, so improved understanding
of the CS-2 freeboard measurement over varying snow and sea ice conditions in open water
areas will be critical to accurately provide sea ice thickness estimates for the Southern Ocean.
Here, we show that modelled snow information has the potential to produce a time series of
snow depth on Antarctic sea ice. However, major developments in modelling capability are
required before their snow products can provide useful information for use in combination with
altimetry data to provide Antarctic sea ice thickness. With improvements to redistribution
mechanisms and adequate representation of the effect of topographic features, atmospheric
models could be used as an alternative to contemporary passive microwave algorithms. Future
work should begin to assess the usefulness of SnowModel products over the larger pack ice
areas, and critically develop a method to (1) incorporate sea ice drift through the atmospheric
model domains, and (2) account for snow loss to leads. If these two influences can be
adequately incorporated, SnowModel could provide a valuable resource for snow and sea ice
thickness investigations over the wider Antarctic sea ice area, especially where snow depth is
high and passive microwave techniques are non-informative.

## 8 Acknowledgments

Gratitude is shown for the support of Antarctica New Zealand and Scott Base staff during the
2011/12 Antarctic field season permitting the collection of *in situ* snow and sea ice
measurements, and the members of field team K053. We thank Glen E. Liston for providing
the code for SnowModel. Further thanks is given to Oliver Marsh and Christian Wild for
productive discussions about the topic. This work was partially supported by NIWA
subcontract C01X1226 (Ross Sea Climate and Ecosystem) and the Marsden Fund Council from
Government funding, managed by Royal Society Te Apārangi. We are grateful to Victoria
Landgraf, Troy Beaumont, and Grant Cottle from Antarctica New Zealand's Scott Base 2011
winter-over team for making the July sea ice thickness measurements as part of the winter
support of a University of Otago Research Grant funded project (PI: Pat Langhorne, AI: Inga
Smith). We thank Peter Green and Inga Smith for their insights into the 2011 sea ice growth
rates, which were supported by the fieldwork and analytical efforts of Greg Leonard, Alex
Gough, Tim Haskell, Pat Langhorne, Jonothan Everts, and by the technical advice of Joe
Trodahl and Daniel Pringle, and the technical support of Myles Thayer, Peter Stroud and
Richard Sparrow. A final thanks is given to Eamon Frazer and Pat Langhorne for the time given
to discussions about and analysis of seawater density in the study region. This research was
completed at Gateway Antarctica, University of Canterbury, Christchurch, New Zealand.

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
