# Peer review of "Snow driven uncertainty in CryoSat-2 derived Antarctic sea ice thickness - insights from McMurdo Sound"

_The Cryosphere, 2018_

## Referee Comment (RC1) · Anonymous Referee #1 · 20 Jul 2018

Review of "Snow depth uncertainty and its implications on satellite derived Antarctic sea ice thickness " by Price et al

In this paper, Price et al. use in situ snow depth and ice thickness data from a McMurdo Sound sea ice survey in 2011 to evaluate different snow depth estimates and ice thicknesses derived from CryoSat-2 radar freeboard estimates. Snow depth and sea ice thickness around Antarctica are highly uncertain, so studies exploring these ideas are clearly needed. However, I found the analysis presented in this study to be lacking the rigor and quality expected for a paper published in The Cryosphere. I list my various concerns below, but also generally felt that the analysis was just not presented well enough to provide a useful resource for the ideas the paper was aiming to highlight (i.e. issues of uncertain snow depths, radar penetration, ice thickness estimates etc.)

[Figure]

I leave the publication decision up to the editor, and provide my review of the paper below.

General comments

The presentation of the different snow depths was pretty bad. I'm very surprised you didn't even show maps of the evolution of the snow depth in SnowModel, AMSRE and the ERA-I precip. I really struggled to get a sense of what they were all doing through this accumulation season.

You use passive microwave snow depths, snow depths converted from ERA-I precip, and snow depths from SnowModel. I was pleased to see such a comparison of different approaches, but the use of SnowModel seemed not well justified considering the uncertainty in precipitation over this study region. In the discussion you mention the benefits of having this sophisticated SnowModel framework, but then also highlight that actually just converting ERA-I precip to snow depth gives arguably better results (through comparisons of the means), so how do you reconcile that? I think you needed to do a lot more comparison of available precipitation/snowfall datasets to get a better idea of what the model is actually doing. How do the PWRF and ERA-I precip/snowfall data differ? It also wasn't clear to me if you were using precip or snowfall in SnowModel and ERA-I..

I was also very confused by the SnowModel configuration and components needed to produce snow depths from this model. What is the Noah-LSM and why is this needed? Seems like this is maybe running an entire atmospheric regional model without any real validation, so why not keep it simple and force this model with a reanalysis like ASR, which is based on Polar WRF?

The use of Pd in this study seemed odd to me, and I think is the wrong way of thinking about this problem. The main issue here is that we have a distribution of returns across the snow layer, including likely some return from no penetration (the snow-air interface) to returns from various penetration depths into the snow layer. What you are showing

is a simplification of this high complexity. I get that you need to do something, but how you've presented this was overly simplistic (a fixed value) in my view and needs to be better explained.

I think you need to provide more context for the survey and the snow data that exists around Antarctica. You say snow depth data are lacking but then present this nice in-situ snow depth dataset. Are similar datasets available elsewhere to see how consistent these ideas are in other areas?

Your title needs changing as I don't really think the results here can help us say anything about snow depth uncertainty and satellite derived Antarctic sea ice other than it being a challenging topic!

The satellite data are described and introduced very crudely throughout. You need to provide better a description of these datasets, especially the Envisat data section.

You mention in the discussion (finally!) the issue of initial conditions, but say are hindered by the fact you don't have good freeze-up info at high resolution, but I would think the passive microwave data is fine for this purpose, especially with the ERA-I analysis? You must have some idea of the bias you introduce if you don't start accumulating until the ice fastens, instead of simply forming.. Is the idea that the ice that forms before fastening is all transported northwards and away from the region? Are there no drift products available to understand that?

I think you should compare using meters, not SWE, as that is what is going into the thickness model. You also didn't even say what the in-situ snow density was.

Specific comments

L27 Not sure I agree with the first line of the introduction!

L42 Decadal trends is pushing it considering we have data from 2003. I think you could be more specific here about the relevant altimetry missions from which thickness data is still lacking.

[Figure]

L43 Completed is strange language to use here.

L54 I think what you want to say here is that there is a long, but old, record of in-situ data of Arctic snow depth from which a climatology has been produced.

L58-59 Reword. Passive microwave data of snow depth available over both poles (where we have FYI).

L73 This terminology doesn't make much sense to me. What is sea ice fast-day-zero?!

L78 Maybe say you compare against in-situ data. Assess uncertainty sounds odd.

L99 Virtual weather station?! Is this not simply the location of an overlapping ERA-I grid cell?

L114 I don't get this gridding discussion. Is this true? It's produced at 25 km then down sampled??

L116 You don't need to state the flag number here..

L133 You need to reword this! Strange sentence structure at the start.

L135 Provide a citation to the CS2 L2 data.

L143 Is this max freeboard based on anything? Surprised this is so low..

L155 Reword Beyond Wingham etc..

L172 Which investigations? Are these the same as other altimetry studies?

L173 Need to reword this. What do you mean by when required? When is this required?!

L191 Has it ever been used for Antarctic snow on sea ice?

L241 Are you using snowfall, not precip? Why can't you also comapre this with the precip from pWRF (or ASR as suggested).

L244-245 So this is the location of the only ERA-I grid-cell in the study area?

Figure 2 Pretty unclear figure with no legend and lines that are hard to distinguish.

L315-318 Ok so two provide snow depth and ERA-I is converted using in-situ density. Reword.

Figure 4 I can't even work out what is being shown here.

L424 I'm not sure what you mean here.

L480 Good point, was ice density not measured directly in this study?!

L485 Unclear how these results indicate accurate snow depths. Figure 3 and 4 showed big differences (especially as a percentage), and ERA-I perhaps performing better..?

It would be clearer if you used centimeter units for the snow depth results throughout!

---

## Referee Comment (RC2) · Anonymous Referee #2 · 23 Jul 2018

This paper introduces a novel approach to deriving snow depth on sea ice in McMurdo Sound using a high-resolution model. The model is compared to other snow depth retrieval methods and their impact on CryoSat-2 sea ice thickness estimates is briefly discussed. The study of different snow depth retrievals is thorough and I'm pleased to see new solutions being developed in the Antarctic. However, I have a couple of significant concerns that need be addressed before the paper can be considered for publication.

1. What the authors are producing is unlikely to be true sea ice thickness, but rather some representative parameter. The factors influencing radar penetration and freeboard in the Antarctic are numerous and are still not well understood (the Willatt et al. (2010) paper remains the key study on this subject). This is partly addressed this by

applying different snow penetration depths in their freeboard to thickness conversion, but this solution which will not capture spatial variation in penetration, or temporal variation if the solution were to be used in different months. More transparency is needed that the retrieval of "thickness" is still highly problematic, and there are limitations in this approach. It should be stated early in the manuscript that "freeboard" is radar freeboard rather than ice freeboard, as this may not be obvious to the wide readership that the paper will attract.

2. The comparison of the various CS-2 sea ice thickness results with in situ data is not sufficient to conclude that any of the CS-2 data show good agreement with in situ data (as suggested on P11 final sentence). Figures 4 and 5 and related discussion provide an initial and basic comparison of sea ice thickness results, but can not be considered an evaluation of the product in any way. In general, some clarification is needed for this analysis:

P10 final sentence: Wording suggests that all CS-2 and in situ thicknesses are mass equivalent thickness. However, it doesn't appear this way from Figure 4 which shows in situ measurements in November falling below the mean mass equivalent thickness. It should be clear in the text what thickness is being plotted/compared. If in situ (red) and CS-2 thicknesses are not equivalent than this assessment needs to be repeated.

I assume that July and November in situ thicknesses are spatial means for those months, but it's not stated in the text

Figure 4: The caption gives the first mention of in situ sea ice thickness measurements being taken in July. This should be included briefly in section 2.1, as surely the July and November data are not being averaged over the same area.

Further comments

Title and abstract: The title is too broad – it suggests that the scope and study area of the paper are far wider than what is presented. The abstract also needs to state that

the study was limited to fast ice in McMurdo Sound.

P1 L27: Understanding of what?

P1 L42-43: Move all discussion on snow depth assessments to next paragraph, which addresses it in more detail. Seems out of place here.

P2 L51-53: I disagree with point 2, that the retracking procedure is a principal source of error in thickness estimates via snow. The presence of snow will slow radar propagation but the waveform shape will be dictated by the roughness of the reflecting surface. The principle of retracking is to select a given location on this waveform that corresponds to "the surface" at nadir without knowledge of its exact location. This is why the ESA L2 product is considered radar freeboard rather than ice freeboard. Therefore, it is the assumed radar penetration that contributes to the error (up to the user), rather than the waveform retracking procedure applied.

P2 L62-66: It is not clear from the author's description that the assumption of zero ice freeboard is only applicable to laser altimetry, where the snow surface is believed to be the dominant scattering horizon. There is no evidence for this being true with radar altimetry, which is why no hemisphere-wide Antarctic sea ice thickness results have been published for CS-2.

P2 L74-76: Confusing sentence structure

P2 L78: "CryoSat-2" to "CS-2"

Section 2.1: Please provide comment on how many snow density, ice freeboard and ice thickness measurements were made at each site

Section 2.2: Not all ice comprising the "large areas" will appear on the same day, so how is the exact date of fast-day-zero established?

P4 L1117-118: Provide a brief (just a sentence will do) summary of how gridded snow depth values are calculated from spectral gradient ratio

P4 L135: Define "SIN" for readers who may not be familiar with CS-2 data

P4 L135: "... **radar** freeboard measurements..." Here would be a good place to highlight that freeboard is radar freeboard, rather than sea ice freeboard. Therefore, "thickness" is just a representative parameter rather than true sea ice thickness.

P4 L143: Was 0.5 m chosen from in situ measurements or otherwise?

P4 L145: Again, the authors can't be sure

P5 L157: See comment on P2 L51-53. The suggestion that the retracking procedure itself introduces uncertainty is misleading. The purpose of the ESA produce is to provide range and freeboard to the "surface" at nadir. It is up to the user to decide what that surface is.

P5 equations: I appreciate the authors consideration of differing penetration depths on Antarctic sea ice retrievals. However, a large number of factors influence radar propagation over Antarctic sea ice (flooding, icy layers, depth hoar, snow ice, crust, sea water wicking etc). Which of these factors has the dominant impact on radar reflection will depend on the age and depth of snow on sea ice. Therefore, penetration depth is unlikely to be constant even over relatively small areas and a more representative way to vary penetration would be through varying penetration depth by a percentage of snow depth (say 25%, 50%, 75%). Why did the authors not choose that approach for this study?

Section 3.1: Provides a very nice, clear introduction to SnowModel

Section 3.2: More information required on the use of ERA-Interim reanalysis data. 1.) Is this the total precipitation 2.) Are there any temperature constraints on what falls as snow 3.) Why is evaporation not considered

Figure 4: Make penetration depth labels larger

P14 L405: Specify ICESsat-2 footprint, and CS-2 footprint earlier in the manuscript

P15 L478: Penetration can also vary spatially over small study areas (see Willatt et al., 2010), which is why a percentage penetration factor may be more applicable than fixed depth

P16 L506-507: "at least as reliable" is a strong statement, and not proved in the manuscript, considering the authors did not show overlap of AMSR-E snow depths compared with in situ Conclusion: It would be good to finish with a statement regarding the potential for Antarctic-wide application of SnowModel (and limitations) for sea ice thickness retrievals, as the paper title suggests

---

## Author Response (AR1)

**Author's Response to Referee #1 comments on "Snow depth uncertainty and its implications on satellite derived Antarctic sea ice thickness" - Price et al.**

We thank Referee #1 for their review and detailed comments. Our responses are below after each point made by the referee and are highlighted in bold.

General comments

The presentation of the different snow depths was pretty bad. I'm very surprised you didn't even show maps of the evolution of the snow depth in SnowModel, AMSRE and the ERA-I precip. I really struggled to get a sense of what they were all doing through this accumulation season. You use passive microwave snow depths, snow depths converted from ERA-I precip, and snow depths from SnowModel. I was pleased to see such a comparison of different approaches, but the use of SnowModel seemed not well justified considering the uncertainty in precipitation over this study region. In the discussion you mention the benefits of having this sophisticated SnowModel framework, but then also highlight that actually just converting ERA-I precip to snow depth gives arguably better results (through comparisons of the means), so how do you reconcile that? I think you needed to do a lot more comparison of available precipitation/snowfall datasets to get a better idea of what the model is actually doing. How do the PWRF and ERA-I precip/snowfall data differ? It also wasn't clear to me if you were using precip or snowfall in SnowModel and ERA-I..

**Thank you for this comment, we appreciate that certain aspects need clarifying. The point of this study was not to prove that SnowModel was superior to other snow products in the Antarctic. This was a first attempt using SnowModel over Antarctic sea ice and the point of the investigation was to evaluate its usefulness by comparing it against other readily available snow products and in situ data. One of the problems with current snow products is that their resolution in comparison to the altimeter satellite footprint is too coarse. There are also accuracy issues associated with passive microwave techniques over rough and deformed sea ice, this of course typical of Antarctic pack ice. If a more comparable, higher resolution snow product was available this would be a step in the right direction, helping to facilitate the useful combination of satellite altimeter data and snow information. We do not see how the uncertainty in precipitation has anything to do with the research approach and if anything justifies the assessment of different snow products in the region, especially given the availability of a rare in situ measurement dataset.**

**Although the ERA-Interim reanalysis has provided a good resource for snow on sea ice in this study it doesn't mean this applies across the wider Antarctic. This is discussed in section 6. Also ERA-Interim performs well with one precipitation value for the entire region and it was not possible to segment it by freeze up area. We do not feel there is a need to reconcile the differences between the models in this respect, we can only compare the pros and cons of each and discuss how they could be applied to a larger area. When the mean of Snow Model across the entire study area is used (so actually comparing to ERA-Interim – apples to apples) Snow Model is + 2 cm against in situ, while ERA-Interim is – 1 cm. We do not think these are colossal differences that the comment infers they are. We have added the sentence "The SnowModel mean swe for all areas at the end of the simulation is 2 cm higher than *in situ* swe mean." in section 4 to reiterate this. This point is already in the abstract and discussion.**

We understand the referee's point about not showing maps, but the authors don't think this would provide much additional information to the reader. Given the differing spatial resolution of different snow datasets, maps would not allow visualisation of differences (e.g. ERA-Interim a singular grid cell at 80 km resolution and SnowModel at 200 m resolution) and this is why the authors opted for a time series plot of snow depth for all of the snow products. However, we do see the value in showing the SnowModel swe and in situ swe on a map in November to compare how well SnowModel produces the in situ observed snow distribution pattern. We have added this as Fig. 4 (below) and added text to the results section:

"*This general overestimation is clearly seen in Figure 4. Values in the eastern most section of the sea ice cover in McMurdo Sound, adjacent to Ross Island are in the order of 20 to 45 cm swe. These values are all larger than the highest in situ measured swe of 17.7 cm and for large areas, they are over double the measured value. In the central area of the Sound, modelled swe decreases in agreement with measured swe with 5 in situ sites agreeing within ± 0.5 cm of SnowModel swe (Fig. 3 and Fig. 4). The extremes, where there is a lot of snow and where there is very little snow both seem to be exaggerated by the model.*"

[Figure]

**Figure 4. SnowModel distribution map displayed as swe over McMurdo Sound, (a) fast ice, (b) open water/pack ice, (c) McMurdo Ice Shelf, (d) Ross Island. The model swe distribution is the mean of the simulation over the *in situ* measurement period (25th November-1st December). The *in situ* measurements were converted to swe via the density measured at each site, if no measurement was taken (21 sites) the average *in situ* snow density was used (385 kgm⁻³). *In situ* measurement locations are shown as black circles and are the mean of the 60 snow measurements taken at each site. The circle sizes are weighted for swe to allow visualisation of the decreasing swe distribution from east to west. Elevation contours are spaced at 400 m intervals; Mt Erebus is the dominant topographic feature on Ross Island to the east of the fast ice.**

**Only the temporal differences in precipitation between ERA-Interim and SnowModel can be compared given the low resolution of ERA. These differences can be visualised in Figure 2. The time series shows a gradual increase in ERA-Interim swe as this model includes no redistribution mechanism. SnowModel exhibits both increases and decreases in swe driven by both precipitation and transport respectively. These differences are described in the manuscript.**

**ERA-Interim used precipitation (water equivalent) which is clearly stated in the text. SnowModel was run to produce a swe product and a snow depth. This was not clear in the text and we have clarified this with "*SnowModel outputs snow depth and swe. The model has a varying density over time. The swe output is important as it allows comparison of the model to the other snow products which have different density assumptions.*" at the end of section 3.1.**

I was also very confused by the SnowModel configuration and components needed to produce snow depths from this model. What is the Noah-LSM and why is this needed?

**Noah-LSM is a Land Surface Model and one of the many components of the PolarWRF. The reason that this scheme has been highlighted in this paper is that LSMs are responsible for the near-surface exchanges between atmosphere-cryosphere and other climate sub-systems. Particularly the Noah LSM has adapted specific parameterizations that makes it ideal to be used over Arctic and Antarctic regions with varying sea ice thickness and snow distributions.**

Seems like this is maybe running an entire atmospheric regional model without any real validation, so why not keep it simple and force this model with a reanalysis like ASR, which is based on Polar WRF?

**ASR is only available over the Arctic and to the best knowledge of the authors no equivalent product for Antarctica exists. The WRF and PolarWRF models have been widely verified across the meteorological community and the results have been reflected in many peer reviewed articles, some of which have been cited in this paper. The main challenge with snow verification in this region is the lack of adequate in situ precipitation observations especially over the sea ice. Yet, the validity of the coupled PolarWRF-SnowModel outcomes are reflected in the sensible snow distribution across the area of study.**

The use of Pd in this study seemed odd to me, and I think is the wrong way of thinking about this problem. The main issue here is that we have a distribution of returns across the snow layer, including likely some return from no penetration (the snow-air interface) to returns from various penetration depths into the snow layer. What you are showing is a simplification of this high complexity. I get that you need to do something, but how you've presented this was overly simplistic (a fixed value) in my view and needs to be better explained.

**We understand this is a simplistic approach regarding the interaction of radar energy and the overlying snow pack but with the available information it is justified. Advancing this approach should be the subject of another, far more detailed study focusing on actual CS-2 waveforms over different types of snow and ice types with validation data. This is a future goal of the authors research group.**

**Further, the ESA retracker has already identified a surface, we are not attempting a retracking procedure ourselves in this study. Given that a surface height has already been identified we are simply evaluating which surface best represents the dominant backscattering surface by varying where this surface might be (using a penetration depth which is essentially just changing the thickness equation to account for different elevations above sea level) and comparing the results to the thickness measured in situ.**

**To make this immediately clear to the reader at the end of section 1, we have added the following to the manuscript: "*The interaction of radar energy with the snow pack is highly complex and here we take a simplified approach given the surface height has already been established by the ESA retracking procedure. Given the uncertainty of the position of the retracking point with reference to the height above sea level we alter the proportion of snow and ice in the thickness estimation assuming different penetration depths and compare the inferred CS-2 thicknesses with in situ information.*"**

**And this in section 2.4: "*Given this uncertainty we apply a simple methodology to discover the range of thicknesses as inferred via this CS-2 data.*"**

I think you need to provide more context for the survey and the snow data that exists around Antarctica. You say snow depth data are lacking but then present this nice in-situ snow depth dataset. Are similar datasets available elsewhere to see how consistent these ideas are in other areas?

**This is a first attempt at using SnowModel and a first attempt at combining multiple snow products with CS-2 data in the Antarctic. Therefore, the scope of this study is at the local scale. We understand the title may have been misleading in this respect and thank the reviewer for indicating this. We have made changes to the title and abstract as suggested by both reviewers.**

**We have a detailed knowledge of this region and with validation can develop ideas and methods for the combination of snow products and satellite altimetry data that could then be applied at the regional/hemispheric scale. We are not currently attempting this, these are early days for this work. The authors are unaware of other satellite validation datasets for CS-2 that would be appropriate for use with these snow products and if they are available we don't have collaborations with other partners for access to these datasets.**

Your title needs changing as I don't really think the results here can help us say anything about snow depth uncertainty and satellite derived Antarctic sea ice other than it being a challenging topic!

**Agreed, the title was misleading and has been changed to "*Antarctic fast ice thickness from CryoSat-2 using different snow product information*". We have slightly modified the abstract to better align with the new title.**

The satellite data are described and introduced very crudely throughout. You need to provide better a description of these datasets, especially the Envisat data section.

**Thanks for pointing out this weakness, we have added the following to the Envisat, CryoSat-2 and AMSR-E sections. We think this is enough relevant information for the reader.**

**Envisat:** "*To identify the dates and the pattern in which the sea ice fastens across the study area, we use a string of C-band Advanced Synthetic Aperture Radar (ASAR) images from Envisat acquired in Wide Swath mode with a spatial resolution of 150 x 150 m.*"

**CryoSat-2:** "*CS-2 is a Ku-band (center frequency 13.6 GHz) radar altimeter launched in 2010. Its on-board altimeter has an approximate footprint size of 380 m x 1560 m and samples along-track at 300 m intervals. The instrument has three modes and operates its interferometric (SIN) mode in the coastal Antarctic. This mode uses both of the satellites antennas to identify the location of off-nadir returns accurately. This is not the dedicated sea ice mode but is still suitable for sea ice freeboard retrieval.*"

**AMSR-E (reworded and additional information added):** "*The snow depth product is gridded to a 12.5 x 12.5 km$^2$ polar stereographic projection and reported as a 5-day running mean, that mean inclusive of that day and the prior 4 days. We remove data where ice concentrations are lower than 20%. Gridded snow depth values are calculated using the spectral gradient ratio of the 18.7 and 36.5 GHz vertical polarisation channels. For snow free sea ice the emissivity is similar for both frequencies. Snow depth increases attenuation from scattering but is more pronounced at 36.5 GHz than at 18.7 GHz, resulting in higher brightness temperatures at 18.7 GHz (Comiso et al., 2003, Markus and Cavalieri, 1998). Using coefficients derived from a linear regression of in situ snow depth measurements on microwave data, and a 36.5-18.7 GHz ratio corrected for sea ice concentration, snow depth can be estimated (Comiso et al., 2003).*"

You mention in the discussion (finally!) the issue of initial conditions, but say are hindered by the fact you don't have good freeze-up info at high resolution, but I would think the passive microwave data is fine for this purpose, especially with the ERA-I analysis? You must have some idea of the bias you introduce if you don't start accumulating until the ice fastens, instead of simply forming.. Is the idea that the ice that forms before fastening is all transported northwards and away from the region? Are there no drift products available to understand that?

**Passive microwave data could be used for freeze-up analysis but its resolution (at best AMSR-E sea ice drift at 6.25 km and concentration at 12.5 km) is too low to be used effectively with SnowModel. Passive microwave could be used for AMSR-E and ERA-Interim but given the paper is a comparison of the different snow products it does not make sense to consider earlier snowfall for one product and not the others as is suggested above. We also have other concerns which resulted in us deciding to use freeze-up instead. These concerns are explained in a paragraph that has now been added in section 2.2:**

"*The sea ice freeze-up provides a point from which snow can begin to accumulate on the sea ice surface. Freeze-up could be identified using passive microwave information, but this data*

*does not provide the spatial resolution to segment the sea ice area appropriately for SnowModel's 200 m resolution. Also, snowfall before fastening is subject to uncertainty from floe movement, flooding events and snow loss to leads, three influences on the eventual snow depth that we have no way of accurately monitoring. With these uncertainties, we have selected the sea ice fastening date to begin snow accumulation.*"

**This sentence has also been added to the discussion:**

"*Early snowfall on more dynamic pack ice will also be subject to flooding, sea spray (both likely to result in snow-ice formation) and loss to leads. These uncertainties must all be considered in future work.*"

I think you should compare using meters, not SWE, as that is what is going into the thickness model. You also didn't even say what the in-situ snow density was.

**The authors do not agree that using meters will provide a better comparison as the densities used in SnowModel and those used for ERA-Interim and AMSR-E are different. We reduce snow depth to swe when appropriate to remove the density bias.**

**As mentioned in an earlier response, the following has also been added to section 3.1 to help clarify this:**

"*SnowModel outputs snow depth and swe. The model has a varying density over time. The swe output is important as it allows comparison of the model to the other snow products which have different density assumptions.*"

**AMSR-E provides a snow depth, we convert this to swe using the in situ measured density of 385 kgm$^{-3}$. The in situ measured snow density was mentioned (L173, L244 old manuscript) but not clearly enough, we have revised this and added more detail around all densities in section 2.4.**

Specific comments

L27 Not sure I agree with the first line of the introduction!

**We are not sure why the reviewer does not agree with this statement as it is not specified. The understanding of Antarctic sea ice processes and properties, extent, area, drift and roughness have all been greatly advanced over the satellite era. Some confusion could be introduced by the vague use of '*few decades*'. We also see how disagreement around advancements in satellite technology is justified. To be more specific we have amended the sentence to '*The knowledge of Antarctic sea ice extent, area, drift and roughness have been greatly improved over the last forty years, principally supported by satellite remote sensing.* '**

L42 Decadal trends is pushing it considering we have data from 2003. I think you could be more specific here about the relevant altimetry missions from which thickness data is still lacking.

**This sentence is referring to satellite altimetry information available from 1995 (Giles et al. 2008 – from ERS-2) to the present day (23 years). The authors agree that more advanced altimeters are only available from 2003 (ICESat). However, the Giles paper shows antecedent instruments are useful and work is also being carried out using Envisat altimetry (Paul et al. 2018). In light of this we feel the decadal time frame is justified. We do not feel it is necessary to name individual missions here especially as the relevant literature is cited prior.**

L43 Completed is strange language to use here.

**Agreed. '*have been completed*' amended to '*are available*'.**

L54 I think what you want to say here is that there is a long, but old, record of in-situ data of Arctic snow depth from which a climatology has been produced.

**End of sentence amended to "*longer period than the Antarctic so climatologies can be produced*"**

L58-59 Reword. Passive microwave data of snow depth available over both poles (where we have FYI).

**Sentence amended to "*The research community lacks snow climatology information in the Southern Ocean; to date only AMSR-E passive microwave data have been used in combination with altimetry to estimate sea ice thickness.*"**

L73 This terminology doesn't make much sense to me. What is sea ice fast-day-zero?!

**Fast-day-zero refers to the first day that the ice is identified as having fastened. This terminology is clearly confusing and has now been removed throughout the manuscript. We have amended this sentence to "*With a high-resolution snow accumulation model called SnowModel (Liston and Elder, 2006a) and the use of synthetic aperture radar imagery we are able to establish when the sea ice fastens and accumulate snow from those dates for three areas of fast ice in McMurdo Sound in the south-western Ross Sea.*"**

L78 Maybe say you compare against in-situ data. Assess uncertainty sounds odd.

**Sentence amended to "*With these different snow depth datasets we infer sea ice thickness via freeboard measurements from CryoSat-2 and compare these results with in situ information.*"**

L99 Virtual weather station?! Is this not simply the location of an overlapping ERA-I grid cell?

**This terminology has been amended to "*The position at which ERA-Interim atmospheric reanalysis data are retrieved is identified by the black circle.*"**

**This sentence has also been added in section 3.2 for clarification: "*Splines were used to interpolate to this position from the three-dimensional ERA-Interim grid.*"**

L114 I don't get this gridding discussion. Is this true? It's produced at 25 km then down sampled??

**For AMSR-E the spatial resolution at observation frequencies of 18.7GHz and 36.5GHz are reported as 25 km and 15 km respectively. The spatial resolution is variable as determined by the footprint which is influenced by satellite altitude, off-nadir angle and beamwidth (Please see table 2.3-12 below from the JAXA AMSR-E Data Users Handbook below).**

**Table 2.3-12  Beam Width and Footprint**

| Frequency | Beam Width (Nominal) | Footprint (Scanning × Proceeding) | Remarks |
|---|---|---|---|
| 6.925 GHz | 2.2° | 43.2 x 75.4 km | In case of; |
| 10.65 GHz | 1.5° | 29.4 x 51.4 km | Satellite Altitude: 705 km |
| 18.7 GHz | 0.8° | 15.7 x 27.4 km | Earth Radius: 6378 km |
| 23.8 GHz | 0.9° | 18.1 x 31.5 km | |
| 36.5 GHz | 0.4° | 8.2 x 14.4 km | |
| 89 GHz A | 0.2° | 3.7 x 6.5 km | |
| 89 GHz B | 0.2° | 3.5 x 5.9 km | |

**Please refer to the JAXA AMSR-E Data Users Handbook for more detail (http://www.eorc.jaxa.jp/en/hatoyama/amsr-e/amsr-e_handbook_e.pdf)**

**The 25 km to 12.5 km downsizing is described in Worby et al. (2008) - '*Snow depth on sea ice is a standard product of the EOS Aqua Advanced Microwave Scanning Radiometer (AMSR-E) instrument. This represents an average over an area of about 25 × 25 km2, gridded to a 12.5 × 12.5 km2 polar stereographic grid [Comiso et al., 2003].*' But little detail is provided beyond this. Comiso et al. (2003) describe the spatial resolution of AMSR-E with the following table:**

TABLE  I
AMSR-E LEVEL 3 $T_B$ AND SEA ICE DATASETS

| PARAMETER | APPROX. RESOL. | GRID RESOL. SIZE | PRODUCT FREQUENCY |
|---|---|---|---|
| TB  (6.9 GHz) | 58 km | 25.0 km | Daily Asc., Desc., & Ave. |
| TB  (10.7 GHz) | 37 km | 25.0 km | Daily Asc., Desc., & Ave. |
| TB  (18.7 GHz) | 21 km | 25.0, 12.5 km | Daily Asc., Desc., & Ave. |
| TB  (23.8 GHz) | 21 km | 25.0, 12.5 km | Daily Asc., Desc., & Ave. |
| TB  (36.5 GHz) | 11 km | 25.0, 12.5 km | Daily Asc., Desc., & Ave. |
| TB  (89.0 GHz) | 5  km | 25.0, 12.5, 6.25 km | Daily Asc., Desc., & Ave. |
| Sea Ice Conc. (%) | | 25.0, 12.5 km | Daily Asc., Desc., & Ave. |
| Sea Ice Temp. (K) | | 25.0 km | Daily Asc., Desc., & Ave. |
| Snow Depth (cm) | | 12.5 km | 5-day average |

**The snow depth derived via the equations in Comiso et al. (2003) combine brightness temperatures from the 18.7 and 36.5 channels at 21 km and 11 km resolution respectively. With the information provided in the JAXA document and supporting literature there must be some downscaling or mechanism to combine the 21 km and 11 km data resolution to the 12.5 km x 12.5 km$^2$ grid spacing. We do not feel this detail is required here as the grid cell size is all that is relevant to the reader to understand our analysis. The AMSR-E section has been reorganised (see amended manuscript) and we have simplified the resolution sentence to: "The snow depth product is gridded to a 12.5 x 12.5 km$^2$ polar stereographic projection and reported as a 5-day running mean, that mean inclusive of that day and the prior 4 days."**

L116 You don't need to state the flag number here..

**Agreed, not immediately relevant to the reader. The reference to flag number has been removed.**

L133 You need to reword this! Strange sentence structure at the start.

**Sentence amended to: "CS-2 was launched in 2010 and houses a Ku-band radar altimeter (centre frequency 13.6 GHz)."**

L135 Provide a citation to the CS2 L2 data.

**Citation not appropriate but we have added a URL for the CS-2 data after 'SIR_SIN_L2' "- available at: http://science-pds.cryosat.esa.int/"**

L143 Is this max freeboard based on anything? Surprised this is so low..

**Yes, this is based on in situ measurements in 2011. The largest measured total freeboard (ice-plus-snow) was 0.46 m.**

L155 Reword Beyond Wingham etc..

Amended to: **"Wingham et al. (2006) indicate the snow-ice interface is represented by the ESA retracked height. No other information is available about the assumptions made here, only that for diffuse echoes in SAR processing, for baseline C, a new retracker was implemented (Bouffard, 2015)."**

L172 Which investigations? Are these the same as other altimetry studies?

**We have added more detail about our density values and amended this section to:**
**"where $\rho w$ (1027 kgm$^{-3}$), $\rho i$ (925 kgm$^{-3}$) and $\rho s$ (385 kgm$^{-3}$) are the densities of water, sea ice and snow respectively. $\rho w$ is informed by an unpublished time series of surface salinity measurements taken from October 2008 to October 2009 along the front of the McMurdo Ice Shelf. The range in pw during this period is less than 1 kgm$^{-3}$. The $\rho i$ value used here is in the middle of the measured range in McMurdo Sound, the use of which is discussed in Price et al. (2014). $\rho s$ is the mean of snow pit measurements at 18 of the in situ measurement sites in 2011."**

L173 Need to reword this. What do you mean by when required? When is this required?!

**This refers to the correction for the speed of light in snow. This only applies when snow is present and/or some penetration is assumed. If no snow is present the correction is not applied and equally if the air-snow interface is assumed i.e. *Pd* = 0 then no correction is applied.**

**We have amended this sentence to: "*When snow is present and penetration is assumed (i.e. Pd > 0), reduction of the speed of the radar wave through the snow pack is corrected following the procedure described in Kurtz et al (2014).*"**

L191 Has it ever been used for Antarctic snow on sea ice?

**No and this is mentioned in the sentence beginning L235 (old manuscript). We have moved this sentence earlier to L233 (new manuscript) as it establishes this earlier to avoid confusion.**

L241 Aren't you using snowfall, not precip? Why can't you also comapre this with the precip from pWRF (or ASR as suggested)

**No the original data from ERA-Interim is precipitation water equivalent. We convert this to snow depth with the average snow density measured in situ. The objective here is to advance PWRF with SnowModel, so simply comparing precipitation from PWRF would not advance the study.**

L244-245 So this is the location of the only ERA-I grid-cell in the study area?

**ERA-Interim is only available on a 0.75° x 0.75° grid which results in an approximate spatial resolution of 80 km. This is larger than the sea ice area in question and is too coarse to resolve at a higher resolution in the study area. Data was extracted from the reanalysis product at 77.7S 165.8E via a spline interpolation and more detail has been included about this in section 3.2 – "*Splines were used to interpolate to this position from the three-dimensional ERA-Interim grid*". Therefore, the resultant ERA-Interim value at this position represents the spline interpolation value with respect to the local ERA-Interim grid cells.**

Figure 2 Pretty unclear figure with no legend and lines that are hard to distinguish.

**We are not sure why the reviewer finds this plot unclear. It is a time series of swe for each of the products with clearly distinguishable lines. The figure caption describes these lines.**

L315-318 Ok so two provide snow depth and ERA-I is converted using in-situ density. Reword.

**Amended to "*Snow depths for each CS-2 freeboard measurement are retrieved from the SnowModel and AMSR-E products directly, while ERA-Interim swe is converted to snow depth using the mean in situ measured density.*"**

Figure 4 I can't even work out what is being shown here.

**This is a key figure to the paper and shows the range in inferred sea ice thickness assuming different heights represented by the ESA retracker between the air-snow and snow-ice interface. These varying penetration depths are plotted for each snow product. We have moved the penetration depth information (*Pd*) off the plot for clarity and attempted to provide the reader with better information about the figure by reworking the text. We have reworded section 5 to try and make it clearer what is going on in this Figure (please see in new manuscript).**

L424 I'm not sure what you mean here.

**This sentence is referring to the fact it is a perfect study area for AMSR-E with no interference from open water, or leads and that most of the fast ice is flat. However, this sentence isn't really necessary and has been removed.**

L480 Good point, was ice density not measured directly in this study?!

**Yes, the value we have used for sea ice density has been measured and is supported by the literature and previous work by the lead author of this work (expanded upon section 2.4 – L210 – new manuscript). The sentence in the discussion is referring to the fact that if the sea ice density was varied through a given range it would change the sea ice thickness estimated from CS-2 freeboard. We have decided to ignore this in this investigation as we are focusing on the combination of snow depth information and altimeter data, but mention it here as it is an additional source of uncertainty in the eventual sea ice thickness estimate.**

**The sentence in the discussion has been amended to: "*As this analysis was focused on the combination of independent snow products and CS-2 altimeter data, the range in sea ice density has not been taken into account. We have confidence in the middle ground ρi value used from previous work in McMurdo Sound (Price et al., 2014).*"**

L485 Unclear how these results indicate accurate snow depths.

**This sentence states that "*The snow distribution from SnowModel accurately captures the measured distribution in November 2011 and produces a swe mean value that is 0.02 m above the mean of in situ validation, but when sea ice is segmented by fastening date large deviations of up to 0.05 m are present in the east where the model has overestimated snow depth.*"**

**'*Accurately*' only refers to the snow distribution produced by SnowModel which is correct. We then directly state actual means for comparisons after this statement. We have clarified this accurate representation of the snow distribution and other concerns with the addition of a snow distribution map – see response to first 'General comment'. We have added a map as Fig.4 to help visualise this.**

Figure 3 and 4 showed big differences (especially as a percentage), and ERA-I perhaps performing better..?

**SnowModel does exhibit large differences from in situ measurements and the entire snow data set is biased high, likely driven by too much accumulation in the east driven by the topographic barrier of Ross Island in the model. This captures reality but the result is exaggerated in the model. This is all stated in the text. The focus of this paper was to compare SnowModel against existing snow datasets, not to prove it was better.**

**Also to reiterate when the SnowModel mean across all areas is taken it is only 2 cm higher than the in situ mean (ERA-Interim) is 1 cm lower. These are not huge differences and do not justify the elevation of ERA-Interim as a superior product.**

It would be clearer if you used centimeter units for the snow depth results throughout!

**We have changed the units to centimeters for all snow plots in the manuscript. Sea ice thickness, related plots and references in the text remain in meters.**

**We disagree with the reviewer which could be based on a misunderstanding. We attempt to clarify below:**

**We have taken the retracked elevation provided by ESA which provides us with surface height. To assess the range in inferred thickness, as a result of the uncertainty surrounding what surface the radar freeboard represents, we used different assumptions about where this surface is and vary these assumptions between the air-snow and snow-ice interfaces. Our analysis (Fig. 5 (in new manuscript)) shows this range of thicknesses**

as trends through the year and essentially shows the uncertainty in the CS-2 inferred thickness from the ESA product from both the freeboard uncertainty and a snow product uncertainty. We argue that this is an evaluation of a sea ice thickness product as derived from CS-2 measurements. The study does not attempt to build a retracking procedure of its own, but is simply using the information available from the ESA L2 product. As there is no more information on what the radar freeboard represents, there is nothing else that can be done. However, by varying the penetration depth (or the range of surfaces the ESA retracked heights could represent) we find using the interpolated in situ dataset (the best available 'snow product') that CS-2 derived thickness, assuming a penetration of 0.07 m, agrees with in situ mass equivalent thickness within 0.02 m. This is a very good result, but we are not suggesting that this can be universally applied but do argue that it is an evaluation of the ESA L2 product. We are open to further discussion about what else the reviewer suggests can be done with the L2 product, or why it is not considered an evaluation of the L2 height product.

P10 final sentence: Wording suggests that all CS-2 and in situ thicknesses are mass equivalent thickness. However, it doesn't appear this way from Figure 4 which shows in situ measurements in November falling below the mean mass equivalent thickness. It should be clear in the text what thickness is being plotted/compared. If in situ (red) and CS-2 thicknesses are not equivalent than this assessment needs to be repeated.

The referee is correct to identify the difference between these two different thicknesses. In Figure 4 both these types of thicknesses are plotted, the mass-equivalent thickness (inclusive of the influence of the sub-ice platelet layer) represented by the black plus-sign and actual solid sea ice thickness represented by the red line. This red line is a linear fit between measurements taken of solid sea ice thickness in July and November. This will be thinner than mass-equivalent thickness as it does not account for the sub-ice platelet layer and is a direct measure using a tape measure of the consolidated solid sea ice. CS-2 thickness is only comparable to mass-equivalent thickness (black plus-sign) as the freeboard measurement will be influenced by the buoyant force of the sub-ice platelet layer (Price et al. 2014). Therefore, for comparison of in situ thickness to CS-2 thickness the reader should only take note of the difference between the black plus-sign and different CS-2 thickness with varying *Pd* values. The red line is an additional resource to give the reader an idea of the expected sea ice growth rate as an additional comparison to the CS-2 thickness trends.

We have made the following changes to give clarity on the different thicknesses:

End of the first paragraph in section 5: "*The only exception to this is the red line in Fig.4 which is a linear fit between two measurements of consolidated sea ice thickness in July and November 2011 used here to show an expected sea ice thickness growth rate for comparison to CS-2 thickness trends.*"

Figure 4 caption, sentences amended to: "*The red line shows sea ice thickness from in situ measurements of consolidated sea ice thickness with a tape measure taken in July and November in one location in the south of McMurdo Sound joined assuming a constant growth rate. The black plus sign is the mean 'mass-equivalent thickness' from all in situ measurements in November. This is slightly thicker than the end of season thickness indicated by the red line given it takes account of the influence of the sub-ice platelet too.*

*This is what CS-2 derived thickness should be compared to as the freeboard measurement from the satellite will also be affected by the buoyant influenced of the sub-ice platelet layer.*"

I assume that July and November in situ thicknesses are spatial means for those months, but it's not stated in the text.

**This needed clarification. The line is simply a linear fit between measured sea ice thickness in July and measured sea ice thickness at one location in McMurdo Sound. This has been used in the plot to show the growth rate during this period for comparison to the CS-2 thickness trends. This has been clarified with the amendments made in response to the previous comment.**

Figure 4: The caption gives the first mention of in situ sea ice thickness measurements being taken in July. This should be included briefly in section 2.1, as surely the July and November data are not being averaged over the same area.

**In addition to the amendments above this has been included in section 2.1: "*Two more in situ measurements of sea ice thickness are included in the analysis. These are two measurements taken at one location in McMurdo Sound in July and November. Assuming a constant growth rate between these measurements they are used in section 5 as a comparison to CS-2 inferred sea ice growth rates. More detail on how the in situ thickness measurements are used and how they should be interpreted is provided in section 5. *"**

Further comments:

Title and abstract: The title is too broad – it suggests that the scope and study area of the paper are far wider than what is presented. The abstract also needs to state that the study was limited to fast ice in McMurdo Sound.

**We agree the title is too broad and have changed it to:**

'*Antarctic fast ice thickness from CryoSat-2 using different snow product information*'

**McMurdo Sound and fast ice are now included in the abstract.**

P1 L27: Understanding of what?

**This sentence has been amended to: "*The knowledge of Antarctic sea ice extent, area, drift and roughness have been greatly improved over the last forty years, principally supported by satellite remote sensing.*"**

P1 L42-43: Move all discussion on snow depth assessments to next paragraph, which addresses it in more detail. Seems out of place here.

**Moved sentence "*Dedicated basin-scale snow depth assessments are available (Markus and Cavalieri, 2006) but continual improvements in our monitoring ability are key to support the current ESA satellite altimeter missions, CryoSat-2 (CS-2) and Sentinel-3 and NASA's planned ICESat-2 expected to be operational in late 2018.*" to next paragraph.**

P2 L51-53: I disagree with point 2, that the retracking procedure is a principal source of error in thickness estimates via snow. The presence of snow will slow radar propagation but the waveform shape will be dictated by the roughness of the reflecting surface. The principle of retracking is to select a given location on this waveform that corresponds to "the surface" at nadir without knowledge of its exact location. This is why the ESA L2 product is considered radar freeboard rather than ice freeboard. Therefore, it is the assumed radar penetration that contributes to the error (up to the user), rather than the waveform retracking procedure applied.

**We agree with the reviewer that this needed re-wording. The sentence has been amended to: "*2. Uncertainty about what surface the retracking point on the radar waveform actually represents between the ice freeboard and snow freeboard. This initial measurement is commonly referred to as radar freeboard.*"**

**We have added a statement and reference directed at surface roughness in section 2.4:**

**"*It is clear that the presence of snow influences the CS-2 height retrieval but precisely how is dependent on the surface roughness (Kurtz et al., 2014; Hendricks et al., 2010; Drinkwater, 1991), its depth (Kwok, 2014) and its dielectric properties (Hallikainen et al., 1986).*"**

**References added for surface roughness inclusion:**

**Hendricks, S, Stenseng, L, Helm, V and Haas, C (2010) Effects of surface roughness on sea ice freeboard retrieval with an Airborne *Ku*-Band SAR radar altimeter. In *International Geoscience and Remote Sensing Symposium (IGARSS 2010), 25–30 July 2010. Proceedings*. Institute of Electrical and Electronics Engineers, Piscataway, NJ, 3126–3129. doi: 10.1109/IGARSS.2010.5654350.**

**Drinkwater, M. (1991) *Ku*-band airborne radar altimeter observations of marginal sea ice during the 1984 Marginal Ice Zone Experiment. *J. Geophys. Res.*, 96(C3), 4555–4572, doi: doi.org/10.1029/90JC01954.**

**The reviewers comment supports the method of the paper. As "the surface" at nadir is undefined in the ESA product (i.e. no indication is given of where it might be) and it is essentially left to the user. The only way for the user to establish what it represents is to compare it against in situ measured freeboard and thickness. In the manuscript we alter the possible positions at which it could be between the air-snow and ice-snow interface and estimate thickness accordingly. This analysis presents the range of uncertainty as presented by the ambiguity of the radar freeboard and the current inability to accurately define at what point above sea level the retracked surface height represents.**

P2 L62-66: It is not clear from the author's description that the assumption of zero ice freeboard is only applicable to laser altimetry, where the snow surface is believed to be the dominant scattering horizon. There is no evidence for this being true with radar altimetry, which is why no hemisphere-wide Antarctic sea ice thickness results have been published for CS-2.

**Good point, "*Using laser altimetry*" added at the beginning of the sentence.**

P2 L74-76: Confusing sentence structure.

**Sentence amended to: "*The high-resolution model results are compared to snow products from two other independent datasets, the first ERA-Interim precipitation and the second satellite passive microwave snow depth from AMSR-E.*"**

P2 L78: "CryoSat-2" to "CS-2"

**Amended.**

Section 2.1: Please provide comment on how many snow density, ice freeboard and ice thickness measurements were made at each site

**Sentence amended to: "*This involved sea ice thickness, freeboard and snow depth/snow density measurements at 39 sites. Freeboard was measured 5 times in a cross profile at each site, once at the centre of the cross and once at the terminus of each line, as was thickness. Mean snow depths for each in situ site represent 60 individual snow depth measurements over that same cross-profile at 0.5 m intervals. Snow density was measured at 18 sites, well distributed across the area, the mean of these sites is used for this analysis. A full overview of the measurement procedure is provided in Price et al. (2014).*"**

Section 2.2: Not all ice comprising the "large areas" will appear on the same day, so how is the exact date of fast-day-zero established?

**We have added this sentence to provide more detail in section 2.2: "*By comparing motion and patterns between sequential images we are able to identify three areas that froze independently of one another.*"**

**The reviewer is right to question the accuracy of the fastening dates and we have addressed this by including this in section 2.2 "*The largest gap in the Envisat image string is 8 days but no large gaps are around key fastening dates. The typical spacing is 1-2 days so we have confidence we have reduced our error in the fastening date to less than 2 days.*"**

**The term fast-day-zero has been removed from the paper as this just caused confusion.**

P4 L1117-118: Provide a brief (just a sentence will do) summary of how gridded snow depth values are calculated from spectral gradient ratio

**More detail provided and this section has been reworded as: "*Gridded snow depth values are calculated using the spectral gradient ratio of the 18.7 and 36.5 GHz vertical polarisation channels. For snow free sea ice the emissivity is similar for both frequencies. Snow depth increases attenuation from scattering and it is greater at 36.5 GHz than at 18.7 GHz, resulting in increased brightness temperatures at 18.7 GHz (Comiso et al., 2003, Markus and Cavalieri, 1998). Using coefficients derived from a linear regression of in situ snow depth measurements on microwave data and a 36.5-18.7 GHz ratio corrected for sea ice concentration snow depth can be estimated (Comiso et al., 2003).*"**

P4 L135: Define "SIN" for readers who may not be familiar with CS-2 data

**Sentence amended and new sentence added: "*The instrument has three modes and operates its interferometric (SIN) mode in the coastal Antarctic. This mode uses both of the satellites antennas to identify the location of off-nadir returns accurately.*"**

P4 L135: ". . . **radar** freeboard measurements. . ." Here would be a good place to highlight that freeboard is radar freeboard, rather than sea ice freeboard. Therefore, "thickness" is just a representative parameter rather than true sea ice thickness.

**We have provided more clarity on exactly what is being represented by our references to freeboard through the paper. Here we have started with: "*The ESA L2 baseline C SIN mode (SIR_SIN_L2 – available at: http://science-pds.cryosat.esa.int/) data set provides a retracked height for the surface over sea ice and this initial measurement is termed radar freeboard.*"**

**Following this in the same section (2.4):**

**"*Each CS-2 radar freeboard measurement is cross-referenced to freeze-up areas 1, 2 and 3 and assigned a snow depth (Ts) value from the described snow products.*"**

**"*Given this uncertainty we apply a simple methodology to discover the range of thicknesses as inferred via this CS-2 data. We explore this possible range by using a varying penetration depth (Pd) into the snowpack. Equation 1 assumes that the snow surface is detected, equation 2 that the sea ice surface is detected and equation 3 that an arbitrary surface at incremental Pd values into the snow pack represents the retracking point varying from 0.02 m to 0.50 m (or to the snow-ice interface, whichever criteria is met first). The radar freeboard is corrected when snow is present and penetration is assumed (i.e. Pd > 0) for the reduction of the speed of the radar wave through the snow pack following the procedure described in Kurtz et al (2014). We derive sea ice thickness (Ti) using the newly corrected freeboard (Fb) and the described equations;*"**

**All the following references to freeboard follow the same logic i.e. they are corrected radar freeboard (*Fb*).**

**It should also be noted that 'true sea ice thickness' is essentially always a representative parameter (within the error of all inputs) from altimetry. This is absolutely the case if the mean backscattering horizon represented by the radar freeboard is unknown. That is the main purpose for trying different horizons in this paper and establishing the range in this representative parameter of sea ice thickness.**

P4 L143: Was 0.5 m chosen from in situ measurements or otherwise?

**Yes, it was selected from in situ information in 2011. "*(as measured in situ in 2011)*" added to the end of this sentence.**

P4 L145: Again, the authors can't be sure

**We have included a description of what we mean by freeboard earlier in the paper as suggested in the comment above (comment P4 L135). What freeboard is in the paper is now established throughout.**

P5 L157: See comment on P2 L51-53. The suggestion that the retracking procedure itself introduces uncertainty is misleading. The purpose of the ESA product is to provide range and freeboard to the "surface" at nadir. It is up to the user to decide what that surface is.

**See response to P2 L51-53.**

P5 equations: I appreciate the authors consideration of differing penetration depths on Antarctic sea ice retrievals. However, a large number of factors influence radar propagation over Antarctic sea ice (ïnˇCooding, icy layers, depth hoar, snow ice, crust, sea´water wicking etc). Which of these factors has the dominant impact on radar reflection will depend on the age and depth of snow on sea ice. Therefore, penetration depth is unlikely to be constant even over relatively small areas and a more representative way to vary penetration would be through varying penetration depth by a percentage of snow depth (say 25%, 50%, 75%). Why did the authors not choose that approach for this study?

**If we have understood the question correctly, we absolutely agree with the reviewer about the complexity of radar interaction and that penetration depth will be variable, even over small areas. However, we do not agree with the proposed percentage approach.**

**If we take a given percentage then penetration universally increases into the snowpack with increasing snow thickness. This is contrary to evidence in the literature, especially if the surface roughness and complexity of the snow structure are increasing over time (i.e. grain size, layering). We could vary the penetration through the season, starting at 75% and decreasing toward 25% at the end of the season but we have no data to support at what rate these percentages should decrease, so we choose a fixed depth. There are pros and cons to each approach but we think a percentage approach could actually introduce additional uncertainty, whereas a fixed depth gives us the range in sea ice thickness through the growth season through many of the potential horizons.**

**In further support of our approach our observations show that the snow surface and volume in the area of investigation are very uniform. We therefore believe that the radar penetration over larger areas is relatively constant (certainly in comparison to pack ice) and given the high latitude and non-summer months included in the analysis maybe even shows little change over time. We appreciate the need for more comprehensive interpretation of the waveform and the use of detailed statistics from the ground to aid this procedure, but that is beyond the scope of this study and requires better data to inform the procedure. Surface roughness information at the radar wavelength scale is required (radiometric roughness) and larger surface roughness features (geometric roughness) along with detailed information on snow depth, layering, grain size and wetness. Ideally this would all be completed and compiled as statistics that represent each CS-2 radar footprint specifically (i.e. georeferenced to the full 380 m x 1560 m footprint).**

**All that the user has to work with from the level 2 product is surface height. The authors do not see the value in attempting to include the complex variables that influence the**

waveform at this stage. This should all be considered when the retracking is done. All we are doing here is varying where the surface height could be between the air-snow and snow-ice interfaces (inclusive). We compare the assumptions to the in situ thickness and establish (i) the range in thickness estimates associated with the uncertainty and user based interpretation of the ESA radar freeboard and (ii) identify at what assumed freeboard interface produces the thickness closest to in situ measured thickness.

Also as the snow depth principally influences the width of the waveform and trailing edge an algorithm based on a percentage of snow depth isn't ideal either (but this isn't even relevant because the retracking is already complete i.e. all we have is a surface height).

With the percentage approach our time series assessment (new Fig. 5) will have a constantly changing penetration which will make the analysis more difficult to interpret.

In section 2.4 we have also added a sentence which clarifies that if the *Pd* is higher than the snow depth, then we assume full penetration to the snow-ice interface as this needed clarification.

"*We explore this possible range by using a varying penetration depth (Pd) into the snowpack. Equation 1 assumes that the snow surface is detected, equation 2 that the sea ice surface is detected and equation 3 that an arbitrary surface at incremental Pd values into the snow pack varying from 0.02 m to 0.50 m (or to snow-ice interface, whichever criteria is met first).*"

Section 3.1: Provides a very nice, clear introduction to SnowModel

**No action taken.**

Section 3.2: More information required on the use of ERA-Interim reanalysis data. 1.) Is this the total precipitation 2.) Are there any temperature constraints on what falls as snow 3.) Why is evaporation not considered

**This is total precipitation reported in mm water equivalent. We have not considered temperature in the ERA-Interim precipitation analysis. From April, at this latitude we have assumed all precipitation is snow. We have not considered evaporation as we expect it to be non-existent from April-November or certainly negligible. We have only observed melting on snow on sea ice in McMurdo in December and this month is not included in the analysis. Sublimation is possible and expected once the sun rises in austral spring (first sunrise 19th August) but we have no way of accurately including this in the snow mass balance analysis for ERA-Interim data.**

Figure 4: Make penetration depth labels larger

**Label sizes increased and also moved off the plot for easier interpretation. Dashed lines also changed to solid line with colour gradient between the two extremes of air-snow and snow-ice interface. See all changes in amended Fig. 5 in the new manuscript.**

P14 L405: Specify ICESsat-2 footprint, and CS-2 footprint earlier in the manuscript

**CS-2 footprint now given in section 2.4.**

**ICESat-2 footprint mentioned in the discussion as it is not relevant when the satellite is described earlier in the manuscript. Sentence in discussion amended to "*These meter-scale features will be important to capture, especially to support compatibility with smaller satellite altimeter footprints, in particularly ICESat-2 with an expected 0.7 m along-track sampling rate (Abdalati et al., 2010).*"**

**Reference added: Abdalati, W., Zwally, H.J., Bindschadler, R., Csatho, B., Farrell, S.L., Fricker, H.A., Harding, D., Kwok, R., Lefsky, M., Markus, T., Marshak, A., Neumann, T., Palm, S., Schutz, B., Smith, B., Spinhirne, J and Webb C. 2010. The ICESat-2 laser altimetry mission, Proceedings of the IEEE, 98(5), 735-751. doi:10.1109/JPROC.2009.2034765.**

P15 L478: Penetration can also vary spatially over small study areas (see Willatt et al., 2010), which is why a percentage penetration factor may be more applicable than fixed depth.

**See response to 'P5 equations' above.**

L506-507: "at least as reliable" is a strong statement, and not proved in the manuscript, considering the authors did not show overlap of AMSR-E snow depths compared with in situ.

**Agreed this statement is too strong, amended to: "*With improvements to redistribution mechanisms and adequate representation of the effect of topographic features atmospheric models could be used as an alternative to contemporary passive microwave algorithms.*"**

Conclusion: It would be good to finish with a statement regarding the potential for Antarctic-wide application of SnowModel (and limitations) for sea ice thickness retrievals, as the paper title suggests.

[revised manuscript text omitted]

**Comment [DP1]:** Title change from 'Snow depth uncertainty and its implications on satellite derived Antarctic sea ice thickness' to 'Antarctic fast ice thickness from CryoSat-2 using different snow product information'

**Comment [DP2]:** Added McMurdo Sound

**Comment [DP3]:** 2011 added.

**Comment [DP4]:** m changed to cm on snow references throughout.

**Comment [DP5]:** Changed from 'large topographic features'

**Comment [DP6]:** Instead of positive bias.

**Comment [DP7]:** Amended from 'the assumptions involved in separating snow and ice freeboard'

**Comment [DP8]:** Sentence amended for clarity.

**Comment [DP9]:** Specifically refer to 0.07 m as opposed to 0.05-0.10 m range. This Pd provides the best agreement with in situ thickness and provides the most useful information to the community.

**Comment [DP10]:** Amended in response to clarity issues highlighted by both reviewers, previously 'The understanding of Antarctic sea ice has greatly improved over the last few decades, 28 principally supported by advancements in satellite capability.'

[revised manuscript text omitted]

**Comment [DP13]:** Simplified from 'permitting the compilation of snowfall climatologies'.

**Comment [DP14]:** Same sentences but shifted around. Sentences referring to snow depth assessments and different altimeter missions have been moved here from the previous paragraph.

**Comment [DP15]:** Added.

**Comment [DP16]:** Modelling and satellite switched around.

**Comment [DP17]:** Added to inform reader it is first use over Antarctic sea ice .

**Comment [DP18]:** Instead of fast-day-zero.

**Comment [DP19]:** Restructured.

**Comment [DP20]:** Added to establish the scope of the paper early. This also addresses Reviewer 2's concerns about neglecting the complexity of radar altimetry in the previous version of the manuscript.

**2 Study area, field and satellite data**

**2.1 McMurdo Sound and field data**

A detailed *in situ* sea ice measurement campaign was carried out in November 2011 on the fast ice in McMurdo Sound (Fig. 1). This involved sea ice thickness, freeboard and snow depth/snow density measurements at 39 sites. Freeboard was measured 5 times in a cross profile at each site, once at the centre of the cross and once at the terminus of each line, as was thickness. Mean snow depths for each *in situ* site represent 60 individual snow depth measurements over that same cross-profile at 50 cm intervals. Snow density was measured at 18 sites, well distributed across the area, the mean of these sites is used for this analysis unless stated otherwise. A full overview of the measurement procedure is provided in Price et al. (2014). Two more *in situ* measurements of sea ice thickness are included in the analysis. These are two measurements taken at one location in McMurdo Sound in July and November. Assuming a constant growth rate between these measurements they are used in section 5 as a comparison to CS-2 inferred sea ice growth rates. More detail on how the *in situ* thickness measurements are used and how they should be interpreted is provided in section 5.

**Figure 1.** McMurdo Sound study area with each freeze-up area as identified by Envisat radar imagery: area 1 – 01/04/2011 (Blue), area 2 – 29/04/2011 (Green), area 3 – 01/06/2011 (Orange) and SnowModel domain bounded by black box. Freeze-up areas are superimposed on a MODIS image acquired on 15 November at the time of maximum fast ice extent in 2011. The locations of 39 measurement sites used to produce the *in situ* snow and sea ice statistics are shown as white triangles. The position at which ERA-Interim atmospheric reanalysis data are retrieved is identified by the black circle.

**Comment [DP21]:** Detail provided on in situ measurements.

**Comment [DP22]:** Additional information.

**Comment [DP23]:** More detail on snow density measurements.

**Comment [DP24]:** Added to provide detail on red line and additional sea ice thickness measurements shown in Fig. 5.

**Comment [DP25]:** Virtual weather station terminology removed and replaced with 'data are retrieved'.

**2.2 Envisat**

The sea ice freeze-up provides a point from which snow can begin to accumulate on the sea ice surface. Freeze-up could be identified using passive microwave information, but this data does not provide the spatial resolution to segment the sea ice area appropriately for SnowModel's 200 m resolution. Also, snowfall, before fastening occurs, is subject to uncertainty from floe movement, flooding events and snow loss to leads, three influences on the eventual snow depth that we have no way of accurately monitoring. With the resolution restriction in mind and these uncertainties, we have selected the sea ice fastening date to begin snow accumulation. To identify the dates and the pattern in which the sea ice fastens across the study area, we use a string of $C$-band Advanced Synthetic Aperture Radar (ASAR) images from Envisat acquired in Wide Swath mode with a spatial resolution of 150 x 150 m. By comparing motion and patterns between sequential images we are able to identify three areas that froze independently of one another. The first area of fast ice was established by 1 April (area 1 – Fig. 1), by the end of April, a second area of fast ice had formed along the southern extremity of the Sound (area 2 – Fig. 1), and by the beginning of June, a third area had fastened (area 3 – Fig. 1). The largest gap in the Envisat image string is 8 days but no large gaps are around key fastening dates. The typical spacing is 1-2 days so we have confidence we have reduced our error in the fastening date to less than 2 days. These three areas persisted for the winter and when combined, made up the fast ice area present in late November when *in situ* measurements were made.

**2.3 AMSR-E**

The EOS Aqua Advanced Microwave Scanning Radiometer (AMSR-E) was operational from December 2002 until 4 October 2011. The snow depth product is gridded to a 12.5 x 12.5 km$^2$ polar stereographic projection and reported as a 5-day running mean, that mean inclusive of that day and the prior 4 days. We remove data where ice concentrations are lower than 20%. Gridded snow depth values are calculated using the spectral gradient ratio of the 18.7 and 36.5 GHz vertical polarisation channels. For snow free sea ice the emissivity is similar for both frequencies. Snow depth increases attenuation from scattering but is more pronounced at 36.5 GHz than at 18.7 GHz, resulting in higher brightness temperatures at 18.7 GHz (Comiso et al., 2003, Markus and Cavalieri, 1998). Using coefficients derived from a linear regression of *in situ* snow depth measurements on microwave data, and a 36.5-18.7 GHz ratio corrected for sea ice concentration, snow depth can be estimated (Comiso et al., 2003). Snow depth retrievals are restricted to dry snow only and to a depth of less than 50 cm. Variable snow properties including snow grain size, snow density and liquid water content influence microwave emissivity from the sea ice surface and the algorithm is reported to have a precision of 5 cm (Comiso et al., 2003) . Given the extreme southern latitude of the study area, snow conditions throughout this study were very dry, supported by snow pit analysis on the sea ice in November with no wet snow or lensing observed. AMSR-E cells are included in the analysis if over 50% of the cell lies within the fast ice mask, and segmented into each freeze up area by that same criteria. 22 AMSR-E cells are used and due to the instrument failure in early October 2011, data for the last two months of this investigation are unavailable.

**Comment [DP26]:** New sentence added.

**Comment [DP27]:** Replaces fast-day-zero.

**Comment [DP28]:** Here we provide a justification for the use of Envisat data as opposed to other data from which we could monitor sea ice freeze up.

**Comment [DP29]:** Envisat description has been given more detail and information is provided about the expected error in the established fastening dates.

**Comment [DP30]:** Description of data simplified for reader, justification provided in the response to the specific comment to the reviewer.

**Comment [DP31]:** More detail provided on AMSR-E snow depth algorithm.

**2.4 CryoSat-2**

[revised manuscript text omitted]

**Comment [DP48]:** Amended to be inclusive of the addition of Fig. 4

**Comment [DP49]:** Added to clearly establish the mean deviation of the entire SnowModel domain over the sea ice from in situ. This was not clear to the reviewers.

**Comment [DP50]:** Simplified from 'Although the model captures the snow distribution on the fast ice'.

**Comment [DP51]:** Figure reference added.

[revised manuscript text omitted]

**Comment [DP57]:** New figure added.

**Comment [DP58]:** Reworded to provide clarity on where from and how snow information is being used in equations 1-3.

**Comment [DP59]:** Sentence added for clarification on what the in situ measurements are used for the red line in Fig. 5.

From equations 1-3, sea ice thickness is highly sensitive to the snow-ice ratio for the measured
freeboard. This results in a large range in sea ice thickness for all snow products through the
growth season (Fig. 5). Using modelled snow depths (Fig. 5a and b) sea ice thickness can vary
by over 2 m from assuming the air-snow interface or snow-ice interface is measured. The
AMSR-E derived thickness trend is not comparable to the model output trends as the last two
months are missing. However, it is useful to highlight the importance of the snow-ice freeboard
ratio. AMSR-E snow depths are high in comparison to the model snow depths at the beginning
of the growth season and they remain relatively stable for its duration. Because of this, the ratio
of ice to snow above the waterline remains very similar. The modelled snow depths gradually
increase and snow makes up an ever increasing proportion of mass above the waterline. If the
air-snow interface is taken to represent $Fb$ then the trend in sea ice thickness through the growth
season is negative for SnowModel and ERA-Interim derived thicknesses. The trend is more
negative for the SnowModel estimate simply because the snow loading is greater. If the snow-
ice interface (equation 2) is assumed to represent $Fb$, thickness trends are too positive. The
mean CS-2 thickness values for November are 2.62 m and 2.77 m for SnowModel and ERA-
Interim respectively compared to an *in situ* thickness of 2.4 m. The trends that result in a
November thickness supported by the *in situ* measurements are those that assume penetration
into the snow cover, analogous with the retracked surface representing a surface between the
air-snow and snow ice interfaces. For thicknesses derived using SnowModel to match *in situ*
thickness a large $Pd$ of 0.5 m is required given the higher snow depth values, while for ERA-
Interim $Pd$ values of 0.1 to 0.15 m place CS-2 thickness estimates closer to *in situ* thickness.

**Comment [DP60]:** More detail provided here about results if a penetration factor is assumed.

The differences in the snow depths from each model result make if difficult to constrain what
$Pd$ value provides CS-2 thicknesses that agree best with measured thickness. To narrow down
the range of most representative $Pd$ values we use interpolated *in situ* measurements for snow
depth as input to the sea ice thickness calculation. We reduce the CS-2 measurements used in
this comparison to the same area bounded by situ measurements. The total range in estimated
sea ice thickness using interpolated *in situ* snow depth between equations 1 and 2 is 1.7 m. For
$Pd$ values 0.02 m through 0.20 m the best agreement between *in situ* thickness and CS-2 derived
thickness is found between 0.05 and 0.10 m (Fig. 6 – third column, 'In situ'). The CS-2
thickness is only 0.02 m thicker than *in situ* thickness for this particular dataset when $Pd = 0.07$
m. The range in SnowModel derived thickness between equations 1 and 2 is nearly 4 m while
the range when using the ERA-Interim data set is almost half that of SnowModel, showing
good agreement with the *in situ* dataset (Fig. 6). Again this large range in thickness reflects the
higher average snow depth produced by SnowModel. The deeper snow creates a larger range
of snow-to-ice ratios.

**Comment [DP61]:** Sentence reworded to help explain logic of using interpolated in situ snow depth as input to CS-2 thickness estimates.

**Comment [DP62]:** Additional sentence added.

[revised manuscript text omitted]

**Comment [DP71]:** Supporting literature has now been added here.

**Comment [DP72]:** Sentence added about requirement for in situ information representative of the actual satellite footprint.

was focused on the combination of independent snow products and CS-2 altimeter data, the
range in sea ice density has not been taken into account. We have confidence in the middle
ground $pi$ value used from previous work in McMurdo Sound (Price et al., 2014) but this is
another source of uncertainty for regional and basin-scale assessments.

**7 Conclusions**

This work has evaluated the ability of three independent techniques to provide snow depth on
fast ice in the coastal Antarctic. The snow distribution from SnowModel accurately captures
the relative distribution measured in November 2011 and produces a swe mean value that is
0.02 m above the mean of *in situ* validation, but when sea ice is segmented by fastening date
large deviations of up to 5 cm are present in the east where the model has overestimated snow
depth. This accurately captures the mechanism of snowfall and transport driven by the
topography of Ross Island, but the rates are higher than in reality. ERA-Interim swe is 1 cm
lower than *in situ* measurements but its coarse resolution prevented the adjustment of
precipitation to sea ice fastening dates. AMSR-E snow depth information suffers from
problems already documented in the literature, and we find that its performance may have again
been influenced by rough sea ice. The snow distribution produced by AMSR-E was opposite
to that provided by SnowModel and measured *in situ* at the end of the growth season. We were
unable to validate the instrument due to its failure 2 months before the *in situ* data was collected.
The uncertainty in the snow depth estimates manifest themselves in the sea ice thickness
estimates from CS-2. A large range in estimated thickness of over 2 m exists if the range in
freeboard used is between the air-snow and snow-ice interfaces. Here, we find CS-2 freeboard
measurements provided by the ESA retracker are most likely representative of a mean
scattering horizon 0.07 m beneath the air-snow interface, in agreement with recent literature.
It is impossible to confidentially constrain this number without reducing uncertainty in the
established sea surface height from which the freeboard is estimated. An improved
understanding of the CS-2 freeboard measurement will be critical to accurately provide sea ice
thickness estimates over varying snow and sea ice conditions in the Southern Ocean.

Here, we show that modelled snow information has the potential to produce a time series of
snow depth on Antarctic sea ice, that could be used with altimetry data to infer sea ice thickness
if the reference surface of the altimeter can be accurately defined. With improvements to
redistribution mechanisms and adequate representation of the effect of topographic features,
atmospheric models could be used as an alternative to contemporary passive microwave
algorithms. Future work should begin to assess the usefulness of SnowModel products over the
larger pack ice areas, and critically develop a method to (1) incorporate sea ice drift through
the atmospheric model domains, and (2) account for snow loss to leads. If these two influences
can be adequately incorporated, SnowModel could provide a valuable resource for snow and
sea ice thickness investigations over the wider Antarctic sea ice area.

**8 Acknowledgments**

Gratitude is shown for the support of Antarctica New Zealand and Scott Base staff during the
2011/12 Antarctic field season permitting the collection of *in situ* snow and sea ice
measurements, and the members of field team K053. We thank Ethan Dale for compiling and
providing ERA-Interim data. Thanks is given to Oliver Marsh and Christian Wild for
productive discussions about the topic. This work was partially supported by NIWA
subcontract C01X1226 (Ross Sea Climate and Ecosystem) and the Marsden Fund Council from
* * *
**Comment [DP73]:** Sentence added about focus on snow products in this study.

**Comment [DP74]:** Number more specific (0.05-0.10 m previously). This is not a change in the analysis we have simply used a more specific number instead of a range.

**Comment [DP75]:** Sentence added about importance of a defined reference surface.

**Comment [DP76]:** Sentence changed from 'at least as reliable as contemporary passive microwave algorithms.' This wording was too strong especially considering untested model application over open ocean.

**Comment [DP77]:** Concluding sentence on application of SnowModel to wider Antarctic sea ice area has been added.

Government funding, managed by Royal Society Te Apārangi. We are grateful to Victoria Landgraf, Troy Beaumont, and Grant Cottle from Antarctica New Zealand's Scott Base 2011 winter-over team for making the July sea ice thickness measurements as part of the winter support of a University of Otago Research Grant funded project (PI: Pat Langhorne, AI: Inga Smith). We thank Peter Green and Inga Smith for their insights into the 2011 sea ice growth rates, which were supported by the fieldwork and analytical efforts of Greg Leonard, Alex Gough, Tim Haskell, Pat Langhorne, Jonothan Everts, and by the technical advice of Joe Trodahl and Daniel Pringle, and the technical support of Myles Thayer, Peter Stroud and Richard Sparrow. A final thanks is given to Eamon Frazer and Pat Langhorne for the time given to discussions about and analysis of seawater density in the study region. This research was completed at Gateway Antarctica, University of Canterbury, Christchurch, New Zealand.

**9 References**

Abdalati, W., Zwally, H.J., Bindschadler, R., Csatho, B., Farrell, S.L., Fricker, H.A., Harding, D., Kwok, R., Lefsky, M., Markus, T., Marshak, A., Neumann, T., Palm, S., Schutz, B., Smith, B., Spinhirne, J and Webb C. 2010. The ICESat-2 laser altimetry mission, Proceedings of the IEEE, 98(5), 735-751. doi:10.1109/JPROC.2009.2034765.

[revised manuscript text omitted]

---

## Editor Decision (ED1)

The author's should consider each of the reviewers' comments when preparing their revised manuscript. However, to help expedite final acceptance, I note the following key points made by the reviewer that should be addressed. I have also added a couple of my own comments on a some points that I think could be made more clear. These additional comments are not meant as reviewer comments, but are easily addressed and would help improve the paper.

Both referees commented on the limited spatial range of the comparison, and very limited comparisons that can be made (only one point for ERA-I, and a few for AMSR-E). I agree with the reviewers that more discussion/qualification of the results with respect to the very limited comparison and atypical conditions for sea ice needs to be included.

Title: Agree with reviewer #1, this is a specific region, and so the title should reflect that and not generalize.

Initial conditions – As pointed out, the fastening date is not necessarily the onset of snow accumulation. The method used here could lead to some underestimation if snow had already accumulated – can you estimate how much it might have influenced results (though, SnowModel is biased high)?

I agree with the reviewer here wrt lines 454-456. While SnowModel as set up may require 200m resolution, you are not comparing at that resolution except with the in situ data in figure 4. At least based on your results, ERA-I does arguably better for CS-2 ice thickness, as reviewer #1 states (error range is lower in Figure 6) . So the value demonstrated by SnowModel here appears to be in matching the spatial pattern of snow distribution, and as you discuss based on physical reasons one would expect SnowModel to be better. But in the manuscript at least, there isn't evidence that SnowModel improves CS-2 thickness estimates (see also comment below on Polar-WRF). You should be clear in your discussion what your results demonstrate, and what they do not.

Figure 2 – agree with reviewer #1, the dots are hard to see.

Accuracy of results –I agree that the qualitative comparison between SnowModel and Figure 4 could be more informative if made quantitatively.

**Additional comments:**

Line 412-414 –It is worth clarifying that the difference in Pd here is because of the differences in predicted snow depth, not something physical, so what Pd does in the case of an incorrect snow depth is to compensate for getting the snow depth wrong, and does not necessarily indicate what the penetration depth really is. For example, equation 3 can be set up for a case where you have a true snow depth, and an estimated snow depth with some error. Then if you try to fit to match the ice

thickness, then you get a Pd that is the sum of the true Pd and a correction (Pde) that results from an error in your snow depth estimate (Tse):

Pde = (pw-ps)/pw*(Tse) = 0.625Tse.

So, if you have overestimated your snow depth, your apparent penetration depth is corresponding larger than the true one, and vice versa.

Pd=0.5m seems too high, given Figure 2 shows a mean snow depth of ~0.1m swe (i.e. ~0.3m actual).  How can you have Pd=0.5m in this case?  You do say you cut off Pd at the snow depth, but I don't see any evidence that 0.5m is correct for any of your products or in situ data. The correction above would imply you'd need to be off by 0.8 m in Ts, which seems implausible.

I think it is important you clarify what is going on here, and be clear that these Pd values you calculate are not necessarily indicative of what is actually happening with the radar reflection.  Your conclusions do properly reflect this and rightly only give the value based on the in situ comparison.

One thing you did not point out is that SnowModel takes as its input precipitation from Polar-WRF, which will be different from ERA-I. So the comparison between ERA-I and SnowModel and in situ (at least for the CS-2 comparison) mostly just shows that the retrieval is sensitive to errors in snow depth, and not which method is necessarily better (SnowModel would presumably be better where, as you note, snow redistribution matters, but you have not shown that this is a factor here).

476-478  - Note that while shot separation for ICESat-2 is 0.7m, you won't get a sufficient number of photons to get a reliable elevation until you sample something like 100 shots. It is unlikely you will be able to resolve meter-scale features. Might be better here to say that you might want to resolve a statistical distribution of features to capture snow accumulation rates in the presence of blowing snow. (not essential, but you might pick a more recent reference for ICESat-2 here, e.g. Markus et al., 2017).

Also note that different retrackers pick different interface positions.  This introduces an error in addition to Pd and snow depth estimation error that could be mentioned.

For figure 5, you match in part based on the slopes.  But I believe the thought behind incomplete penetration into the snowpack is due to some physical scattering horizon, either an icy layer or perhaps wicked brine. Then could it be that Pd is at different depths at different times?

**Minor/technical points:**

Line 61 – ICESat-2 has successfully launched now, so this statement should be updated.

Section 2.3 – as pointed out by the reviewer, it isn't clear if you have calculated snow depth yourself or used an existing product. If the latter, the dataset used should be referenced.

Line 158 – should be "in coastal Antarctic" I think.

Line 180 – "but precisely how *it* is dependent"

Line 269 – should be "see Hines et al., (2015)"

Figure 4 – you might consider narrowing the scale here, your in situ measurements go up to ~15 cm, but your SnowModel scale goes to 180!  It would be more clear if these scales were similar.

---

## Author Response (AR2)

**Response to second round of reviewer responses for tc-2018-92 "*Snow depth**
***uncertainty and its implications on satellite derived Antarctic sea ice thickness*"**

**The authors would like to thank the reviewers and the editor for the continuation of their detailed**
**comments on the manuscript. We have addressed these below as responses to Reviewer #1,**
**Reviewer #2 and Editors Comments in bold. There are a few key changes to the manuscript**
**highlighted first.**

1.  **The principal change to the manuscript is a change to the ERA-Interim dataset after an**
**error in the original code was identified while re-gridding the ERA-I dataset. The ERA-I**
**dataset was too low by an order of magnitude; the ERA-I accumulation was**
**underestimated in the previous versions of the document. The authors apologise for this**
**error in the initial submission. The change to this result makes SnowModel far superior**
**to ERA-I for its eventual snow depths when compared to in situ measurements in**
**McMurdo Sound. All changes to results are highlighted along with relevant changes to**
**the text. The correct dataset has been offered to the editor and is available for the**
**reviewers to view if required. The old and new Figure 2, which best showcase the**
**difference are provided at the end of this response.**
2.  **We have changed the title to accommodate the reviewers concerns and agree the initial**
**title was too broad. New title: "*Snow driven uncertainty in CryoSat-2 derived Antarctic sea*
*ice thickness - insights from McMurdo Sound*"**
3.  **ERA-Interim in the text has been abbreviated to ERA-I.**

**Reviewer #1**

Response to author comments for the "Snow depth uncertainty and its implications on satellite derived
Antarctic sea ice thickness" paper by Price et al.

I like Figure 4, I'm glad you've included it. However, I still don't buy the decision not to also show the
ERA-I grid-cells. I think there could be several grid cells covering your study area and that might be
telling to assess its performance. I would guess you could even have one or two grid-cells to represent
the three fast ice regions too, making the later analysis a lot more interesting (rather than just showing
the ERA-I line as a study region mean).

**The reviewer was correct to identify that multiple ERA-I grid cells covered the study region, at**
**the latitude of the study the 0.75º x 0.75º cell size resulted in 14 separate ERA-I cells in the**
**snowmodel domain. The central point of these cells are now displayed in Figure 1. Because of this**
**we have now included a 10 x 10 ERA-I grid as the basis of the analysis which is shown in the**
**results as Figure 4b. The study region is now segmented for fastening date using the ERA-I 10 x**
**10 grid. However, all changes to the results are principally driven by the major correction to the**
**ERA-I dataset as indicated above.**

I'm also still confused about some of the snowmodel choices - can it be run for coarser resolutions? If
so, why not do that? What benefit is there for running it at 200 m? My main point really is that the mean
precip biases might be more important than capturing the high spatial variability.

**In the modeling space when it comes to wind (given that snow distribution was the major**
**objectives) we try to go as high-res as possible. We're looking at the complexity of topography**
**and decide which resolution can capture most of the orographical detail. At the same time, the in**
**situ measurements highlighted the role of topographical complexity in snow distributions.**
**Improving the wind direction and speed is one of the strengths of the snowmodel. Snowmodel**
**cannot achieve this unless it has access to very high-resolution topography.**

**WRF with 3km resolution could not capture all those detailed topographic complexities in the study region. Snowmodel is able to improve the wind speed and direction coming from the coarser WRF outputs but higher-res topography is critical. 200 m was the most detailed topography data we had access to, we would have done it at 10 m if we had access to that data.**

**At scales around 200 m we are also providing snow information at similar spatial scales to the satellite footprint. We understand snowmodel hasn't provided snow depths of the desired accuracy but it is attempting to include more complexity. Future developments will likely improve its performance.**

**We have provided statistics for each of the fastening areas and the entire study area from this higher-res data which addresses the mean precipitation and provides a comparison to the other snow products.**

**Higher resolution snow depth products are also desirable to capture snow information at the same spatial scales as the satellite altimeter footprint.**

"ERA-Interim used precipitation (water equivalent) which is clearly stated in the text. Snowmodel was run to produce a swe product and a snow depth. This was not clear in the text and we have clarified this with "snowmodel outputs snow depth and swe. The model has a varying density over time. The swe output is important as it allows comparison of the model to the other snow products which have different density assumptions." At the end of section 3.1. ":

I think you've missed the point here. ERA-I provides snowfall and total precip as different variables. Why note use the snowfall variable?

**Apologies for the miscommunication, we did not use the ERA-I snow product and have just used the total precipitation variable (swe) as no rainfall is expected at this latitude for the study period. We then use the more accurate density measured in situ to convert precipitation to snow depth when required.**

"Antarctic fast ice thickness from cryosat-2 using different snow product information" I still think this title needs work! Can you reference more directly your study area as again the title is inferring a wider study than what is presented (it's very local scale). E.g. "Comparison of snow depths in mcmurdo Sound from in-situ data and various snow products and its impact on sea ice thickness altimetry"?

**Understood, title amended to:**

**"Snow driven uncertainty in CryoSat-2 derived Antarctic sea ice thickness - insights from McMurdo Sound"**

Extra discussion on satellite data products:

It's still unclear what exact products you are using. Can you provide the links as this may help clarify things (you aren't calculating these data yourself, right?..). E.g. The Envisat description doesn't make this clear. Are you doing this processing or obtaining this information from an existing product?

**We have added additional information about the Envisat processing and the source of the AMSR-E data. Enough information is provided about the products for the reader. The Envisat data link is difficult to include in the text as it involves registration online via ESA. This all gets a little complicated in the text and is not required.**

**Envisat – *"we use a string of C-band Advanced Synthetic Aperture Radar (ASAR) images from***
***Envisat acquired in Wide Swath mode. We process these files using using GAMMA Software to***
***produce ASAR imagery with a spatial resolution of 150 x 150 m".***

**AMSR-E – *"The snow depth product provided by NSIDC***
***(https://nsidc.org/data/AE_SI12/versions/3#) is provided at a 12.5 x 12.5 km2 polar stereographic***
***projection and reported as a 5-day running mean, that mean inclusive of that day and the prior 4***
***days.".***

Comments on initial conditions:

I don't agree with your response to this and your discussion of grid resolutions. You could apply a
constant value and just distribute that over the high resolution snowmodel grid if you want, so doing
this for a 12 km dataset would be possible also. You just aren't capturing the spatial variability. I still
think need to make this potentially missing snow clearer, and hopefully provide some estimate at what
potential bias that might introduce.

**The authors think this response accurately depicts the situation. The additional snow delivered**
**before sea ice fastening will be negligible. Pack ice in mcmurdo Sound is transported north into**
**the wider Ross Sea region, until it fastens it does not remain in the Sound for long.**

**An example of this can be visualised over the study region at the link below:**

**(https://worldview.earthdata.nasa.gov/?P=antarctic&l=VIIRS_SNPP_correctedreflectance_true**
**color(hidden),MODIS_Aqua_correctedreflectance_truecolor(hidden),MODIS_Terra_corrected**
**reflectance_truecolor,Coastlines,AMSRE_Sea_Ice_Concentration_12km(hidden),AMSRE_Sea_**
**Ice_Brightness_Temp_89H(hidden)&t=2011-02-20-T00%3A00%3A00Z&z=3&t1=2011-03-20-**
**T00%3A00%3A00Z&v=68147.42217165558,-1504445.119093321,612172.8978546829,-**
**1117069.1665325488&r=-162.064&ab=off&as=2011-03-27T00%3A00%3A00Z&ae=2011-04-**
**03T00%3A00%3A00Z&av=3&al=true)**

**The fast ice in the Sound fully breaks out during a storm event (21-23 Feb 2011) including sections**
**of the McMurdo Ice Shelf. Sea ice begins and continues to form in the south-western Ross Sea**
**through the first half of March. All this pack ice is forced northward by southerly winds in the**
**wider Ross Sea region. It is not until it fastens that it remains in the model domain and actually**
**accumulates snow. The authors think the fastening date actually provides quite a robust measure**
**of time zero for snow accumulation in this study region. Additionally even if this were not the**
**case, all model snow data sets are biased high and inclusion of this unnecessary factor in this case**
**would increase the discrepancy between model results and in situ measurements.**

**We have added an additional sentence in section 2.2:**

**"In McMurdo Sound during the freeze-up period, pack ice is generally advected north out of the**
**study area unless it fastens."**

Response to SWE units:

OK but I think it will be illuminating to see what the in-situ density and SnowModel densities are.

**Mean in situ density is provided in the text. We have not investigated the additional uncertainty**
**introduced by varying snow density in this study. SnowModel incorporates its varying density**
**through the growth season in the snow depth output. However, we have no information on ERA-**
**I snow density nor AMSR-E and can only reduce to swe via the end of growth season in situ**
**measurements. We choose not to investigate this additional source of uncertainty. To effectively**
**investigate this in situ snow density would need to be collected through the growth season, a**
**significant logistical task. This would allow the correct numbers to be entered monthly with**

**coincident CS-2 measurements. Better constraining these values is important and a part of our**
**future work.**

AMRS-E gridding comment:

Thanks for the information. I think say 'provided at' instead of 'gridded to' as this currently makes it
seem like you do the gridding.

**This sentence has been amended to:**

**"*The snow depth product provided by NSIDC (https://nsidc.org/data/AE_SI12/versions/3#) is***
***provided at a 12.5 x 12.5 km2 polar stereographic projection and reported as a 5-day running mean,***
***that mean inclusive of that day and the prior 4 days.*"**

Cryosat data link:

Thank you for providing this. Links to data are needed.

**Of course. Agreed!**

"ρs is the mean of snow pit measurements at 18 of the in situ measurement sites in 2011.":
Why only 18 of the sites?

**It was only measured at these sites given time constraints during the fieldwork. They have a**
**representative spread across the study area. Sentence amended to "*$\rho_s$ is the mean value taken from***
***18 of the 39 in situ sites where snow density was measured.*"**

"We are not sure why the reviewer finds this plot unclear. It is a time series of swe for each of the
products with clearly distinguishable lines. The figure caption describes these lines. ":

The circles are tiny so this hardly distinguishes it from a solid line. This still needs improvement.

**This figure has been replotted with different symbols.**

New Figure 5 (was Figure 4): I don't understand the use of linear fits here. Does it look too noisy if
you     use     the     actual     values?     How     about     bar     charts     for     the     different     months?
**We have used these here to give the reader a better impression of the growth rates through the**
**season. Yes, it is difficult to interpret with the monthly sea ice thickness means as stand-alone**
**data points. The linear fits clarify this and also let the reader compare the CS-2 data to the in situ**
**measured thicknesses points and line (red) also in the figure. We don't agree bar charts would**
**represent the growth through the season well.**

Comments on accuracy of the results:

How do you judge this to be an accurate spatial distribution? The map gets the broad spatial distribution
pattern correct? If so you could be more explicit. Unsure what you mean by 'correct'. Again, I really
think that despite the coarseness of ERA-I you're domain is big enough to get a few grid-cells that could
provide some assessment of a regional distribution (albeit only with one or two grid cells per region).

We have made this clearer in the text with:

**"*This broad spatial distribution produced by SnowModel compares well with in situ measurements***
***and general observations during fieldwork in November 2011, which recorded an increasing gradient***
***in snow depth from west to east (Fig. 4).*"**

**We have amended the ERA-I analysis with the 10x10 grid to improve the resolution.**

I think you should drop the 0.02 mean bias as this is just because you have compensating errors in your
regional differences. The 0.05 cm differences are ~30-50% off, right? So still pretty big!

**The authors think it is appropriate to use the study area mean as it gives an idea of the total swe**
**delivered to the study region compared to that measured in situ. Yes there are regional**
**differences, these are reported along with the developments required to improve the model.**

If you just compared the ERA-I mean with the in-situ values in November I think you would get similar
errors to the snowmodel values, correct? This would imply snowmodel isn't doing any better than ERA-
I. See earlier comments about ERA-I.

**With the revised ERA-I data set SnowModel and ERA-I results are now significantly different,**
**SnowModel far outperforms ERA-I. This is all appropriately addressed in the text.**

Reviewer #2

The authors have done a good job in addressing my concerns regarding the initial manuscript
submission. I do however have a couple of small remaining issues that should be considered before
publication.

1.) Related to the second significant concern of my first review (that the comparison of the various CS-
2 sea ice thickness results with in situ data), I appreciate the author's clarifications and expansions. I
understand the author's stance that this is in fact an evaluation of the product (and they do not claim it
is a detailed validation). However, I would expect a little further comment within the manuscript on the
spatial limitations of this comparison.

**We feel this is communicated in the manuscript now especially with the amended title. We have**
**also added "*in McMurdo Sound.*" in the abstract.**

2.) L196: Quantify "incremental". Could the authors also justify why they plotted the increments they
did in Figures 4 and 5, considering Pd = 0.07 gives the best agreements between CS-2 and in situ
thickness?

**Sentence amended to:**

**"*Equation 1 assumes that the snow surface is detected, equation 2 that the sea ice surface is detected*
*and equation 3 that an arbitrary surface at varying Pd values into the snow pack (0.02 m, 0.05 m,*
*0.10 m, 0.15 m, 0.30 m and 0.50 m - or to the snow-ice interface, whichever criteria is met first)*
*represents the retracking point.*"**

**The 0.07 m is representative of the in situ interpolated snow data, for the other data sets we chose**
**to display a range of possibilities through the 0.02, 0.05, 0.10, 0.15, 0.30 and 0.50 range. So in other**
**words when the snow information has the least error 0.07 m is the most accurate *Pd*, given the**
**range in snow provided by the other datasets it is necessary to show a larger range. 0.07 m would**
**not produce the best results with the other products.**

**Editor's Comments:**

The author's should consider each of the reviewers' comments when preparing their revised manuscript.
However, to help expedite final acceptance, I note the following key points made by the reviewer that
should be addressed. I have also added a couple of my own comments on a some points that I think could be made more clear. These additional comments are not meant as reviewer comments, but are
easily addressed and would help improve the paper.

Both referees commented on the limited spatial range of the comparison, and very limited comparisons
that can be made (only one point for ERA-I, and a few for AMSR-E). I agree with the reviewers that
more discussion/qualification of the results with respect to the very limited comparison and atypical
conditions for sea ice needs to be included.

**The title has been amended, and McMurdo Sound has been specifically referred to in the abstract**
**for a second time. We have also added to the conclusion and amended some text:**

**"***Sea ice in McMurdo Sound is atypical of Antarctic pack ice, so improved understanding of the CS-*
*2 freeboard measurement over varying snow and sea ice conditions in open water areas will be*
*critical to accurately provide sea ice thickness estimates for the Southern Ocean.***"**

Title: Agree with reviewer #1, this is a specific region, and so the title should reflect that and not
generalize.

**Title has been amended to: "***Snow driven uncertainty in CryoSat-2 derived Antarctic sea ice*
*thickness - insights from McMurdo Sound***"**

**The following sentence has also been added to the introduction:**

**"***The uncertainty associated with these two factors* [points 1 and 2 in the introduction] *has not been*
*directly investigated using satellite altimeter information over Antarctic sea ice. This work provides*
*insights from a case study region, McMurdo Sound Antarctica.***"**

Initial conditions – As pointed out, the fastening date is not necessarily the onset of snow accumulation.
The method used here could lead to some underestimation if snow had already accumulated – can you
estimate how much it might have influenced results (though, snowmodel is biased high)?

**Please see response to reviewer 1's comment above. The authors still support that the fastening**
**date is a good measure to begin accumulation given the routine advection of pack ice north, at**
**least in 2011. As the editor points out even if it were worth including additional snow**
**accumulation days prior to fastening, it would cause a larger deviation of model datasets from in**
**situ information.**

I agree with the reviewer here wrt lines 454-456. While snowmodel as set up may require 200m
resolution, you are not comparing at that resolution except with the in situ data in figure 4. At least
based on your results, ERA-I does arguably better for CS-2 ice thickness, as reviewer #1 states (error
range is lower in Figure 6). So the value demonstrated by snowmodel here appears to be in matching
the spatial pattern of snow distribution, and as you discuss based on physical reasons one would expect
snowmodel to be better. But in the manuscript at least, there isn't evidence that snowmodel improves
CS-2 thickness estimates (see also comment below on Polar-WRF). You should be clear in your
discussion what your results demonstrate, and what they do not.

**This discussion point has changed given the correction to the ERA-I dataset, SnowModel is far**
**superior to ERA-I. We are also using the higher resolution advantage of SnowModel to directly**
**extract snow depth values with the 200 m grid cells for each 380 x 1560 m CS-2 altimeter retrieval.**
**This should be the goal for future missions and modelling efforts to tie together the discrepancies**
**in spatial scales between required data products. The best that is currently achieved by the other**
**snow products in the study is 12 km.**

Figure 2 – agree with reviewer #1, the dots are hard to see.

**This figure has been replotted with different symbols.**

Accuracy of results –I agree that the qualitative comparison between SnowModel and Figure 4 could
be more informative if made quantitatively.

**The aim of Figure 4 is to help the reader visualize the snow distribution as suggested by the**
**reviewers in the previous revision. The authors agree this is important. Figure 3 displays**
**quantitatively the difference between the in situ sites and Snow Model values.**

Additional comments:

Line 412-414 –It is worth clarifying that the difference in Pd here is because of thickness, then you get
a Pd that is the sum of the true Pd and a correction (Pde) that results from an error in your snow depth
estimate (Tse):

$Pde = (pw-ps)/pw*(Tse) = 0.625Tse$.

So, if you have overestimated your snow depth, your apparent penetration depth is corresponding larger
than the true one, and vice versa.

**We have clarified this by adding "*This range in inferred thickness is driven by the amount of snow***
***produced by the models as Eq. 1 and Eq. 2 subtract and add the product of this value in their second***
***terms respectively. As the snow depth increases, in some cases to higher values than the measured***
***freeboard the Pd simply provides a correcting factor for this discrepancy.***"

Pd = 0.5 m seems too high, given Figure 2 shows a mean snow depth of ~0.1m swe (i.e. ~0.3m actual).
How can you have Pd=0.5m in this case? You do say you cut off Pd at the snow depth, but I don't see
any evidence that 0.5m is correct for any of your products or in situ data. The correction above would
imply you'd need to be off by 0.8 m in Ts, which seems implausible.

**Figure 2 shows the mean swe for each fastening area, not the maximum values. Maximum values,**
**especially in the east are far higher (see Figure 4a). Maximum values for swe for SnowModel are**
**in the order of 20-30 cm swe and for ERA-I they are nearly 30 cm swe. This justifies plotting a**
**0.5 m (snow depth) *Pd*. *Pd* is cut off at the snow depth so is only applied when appropriate.**

I think it is important you clarify what is going on here, and be clear that these Pd values you calculate
are not necessarily indicative of what is actually happening with the radar reflection. Your conclusions
do properly reflect this and rightly only give the value based on the in situ comparison.

**Added in abstract:**

**"*Because of this ambiguity we vary the proportion of ice and snow that represents freeboard – a***
***mathematical alteration of the radar penetration into the snow cover and assess this uncertainty in***
***McMurdo Sound.***"**

**Added in section 2.4**

**"*We explore this possible range by changing the amount of snow and ice assumed to represent the***
***freeboard measurement in the thickness equation. There is no physical change to the actual radar***
***penetration, the inferred thickness is simply altered mathematically using a varying penetration***
***depth (Pd) into the snow pack.***"**

**To summarise, we are taking the available snow products and producing one of our own. We**
**combine these with altimetry and are then left with the further uncertainty associated with the**
**mean scattering horizon. The reason for the *Pd* assessment is to explore the range of uncertainty**

**associated with this. There is no other way to show these results until the ambiguity is the CS-2**
**fb is better constrained. From the available data we can only say sea ice thickness is between x**
**and y, and the range is very large.**

One thing you did not point out is that snowmodel takes as its input precipitation from Polar-WRF,
which will be different from ERA-I. So the comparison between ERA-I and snowmodel and in situ (at
least for the CS-2 comparison) mostly just shows that the retrieval is sensitive to errors in snow depth,
and not which method is necessarily better (snowmodel would presumably be better where, as you note,
snow redistribution matters, but you have not shown that this is a factor here).

**It is stated in the manuscript (L244-248 second version, L251-255 latest version) that hourly**
**atmospheric forcing were generated by version 3.5 of the polar-optimized version of the Advanced**
**Research Weather Research and Forecasting Model. The revised ERA-I dataset also now show**
**that ERA-I is not suitable for retrieving sea ice thickness in this region at least.**

476-478 - Note that while shot separation for icesat-2 is 0.7m, you won't get a sufficient number of
photons to get a reliable elevation until you sample something like 100 shots. It is unlikely you will be
able to resolve meter-scale features. Might be better here to say that you might want to resolve a
statistical distribution of features to capture snow accumulation rates in the presence of blowing snow.
(not essential, but you might pick a more recent reference for icesat-2 here, e.g. Markus et al., 2017).

**Removed specific reference "*with an expected 0.7 m along-track sampling rate*" and added Markus**
**et al., 2017 reference.**

Also note that different retrackers pick different interface positions. This introduces an error in addition
to    Pd    and    snow    depth    estimation    error    that    could    be    mentioned.
**Noted, only one retracker is being used here so this error source is absent. I have specifically made**
**comparison between retracking techniques in Price et al. (2015).**

For figure 5, you match in part based on the slopes. But I believe the thought behind incomplete
penetration into the snowpack is due to some physical scattering horizon, either an icy layer or perhaps
wicked brine. Then could it be that Pd is at different depths at different times?

**This is true and this was discussed in the previous response. The snow pack is particularly**
**homogenous in this region so such physical influences on the scattering horizon should be at a**
**minimum. It is also beyond the scope of this study to start trying to improve interpretation of the**
**radar waveform, we are taking an elevation product from the ESA retracker and inferring**
**thickness from those estimates.**

Minor/technical points:

Line 61 – icesat-2 has successfully launched now, so this statement should be updated.

**Corrected and changed to "*and NASA's laser altimeter mission ICESat-2*".**

Section 2.3 – as pointed out by the reviewer, it isn't clear if you have calculated snow depth yourself or
used an existing product. If the latter, the dataset used should be referenced.

**Sentence amended to:**

**"*The snow depth product provided by NSIDC (https://nsidc.org/data/AE_SI12/versions/3#) is***
**provided at a 12.5 x 12.5 km2 polar stereographic projection and reported as a 5-day running mean,**
**that mean inclusive of that day and the prior 4 days.**"

Line 158 – should be "in coastal Antarctic" I think.

**339 Sentence amended to "***The instrument has three modes and over the coastal Antarctic operates its***
**340 *interferometric (SIN) mode.***"

Line 180 – "but precisely how it is dependent"

**342 Commas added so this sentence is read correctly.**

Line 269 – should be "see Hines et al., (2015)"

**344 Amended.**

Figure 4 – you might consider narrowing the scale here, your in situ measurements go up to ~15 cm,
but your snowmodel scale goes to 180! It would be more clear if these scales were similar.

**347 Scale amended.**

[Figure]

[Figure]

**Manuscript mark-up**

Snow driven uncertainty in CryoSat-2 derived Antarctic sea ice thickness -
insights from McMurdo Sound

> **Commented [DP3]:** New title

Daniel Price[1], Iman Soltanzadeh[2] & Wolfgang Rack[1], Ethan Dale[3]

[1]Gateway Antarctica, University of Canterbury, Private Bag 4800, Christchurch, New Zealand
[2]Met Service, 30 Salamanca Road, Kelburn, Wellington, 6012, New Zealand
[3]Department of Physics and Astronomy, University of Canterbury, Christchurch, New Zealand

*Correspondence to*: Daniel Price (daniel.price@canterbury.ac.nz)

> **Commented [DP4]:** New author added

> **Commented [DP5]:** Additional affiliation added for new author

**Abstract.** Knowledge of the snow depth distribution on Antarctic sea ice is poor but is critical to obtaining sea ice thickness from satellite altimetry measurements of freeboard. We examine the usefulness of various snow products to provide snow depth information over Antarctic fast ice in McMurdo Sound with a focus on a novel approach using a high-resolution numerical snow accumulation model (SnowModel). We compare this model to results from ECMWF ERA-Interim precipitation, EOS Aqua AMSR-E passive microwave snow depths and *in situ* measurements at the end of the sea ice growth season in 2011. The fast ice was segmented into three areas by fastening date and the onset of snow accumulation was calibrated to these dates. SnowModel captures the spatial snow distribution gradient in McMurdo Sound and falls within 2 cm snow water equivalent (swe) of *in situ* measurements across the entire study area. However, it exhibits deviations of 5 cm swe from these measurements in the east where the effect of local topographic features has caused an overestimate of snow depth in the model. AMSR-E provides swe values half that of SnowModel for the majority of the sea ice growth season. The coarser resolution ERA-Interim, produces a very high mean swe value 20 cm higher than *in situ* measurements. These various snow datasets and *in situ* information are used to infer sea ice thickness in combination with CryoSat-2 (CS-2) freeboard data. CS-2 is capable of capturing the seasonal trend of sea ice freeboard growth but thickness results are highly dependent on what interface the retracked CS-2 height is assumed to represent. Because of this ambiguity we vary the proportion of ice and snow that represents freeboard – a mathematical alteration of the radar penetration into the snow cover and assess this uncertainty in McMurdo Sound. The range in sea ice thickness uncertainty within these bounds, as means of the entire growth season are 1.08 m, 4.94 m and 1.03 m for SnowModel, ERA-Interim and AMSR-E respectively. Using an interpolated *in situ* snow dataset we find the best agreement between CS-2 derived and *in situ* thickness when this interface is assumed to be 0.07 m below the snow surface.

> **Commented [DP6]:** Main results changed to mean annual range between Pd extremes to better express the uncertainty.

> **Commented [DP7]:** Reworded

[revised manuscript text omitted]

than swe for SnowModel for areas 2 and 3 after the first large increase in swe in these areas in
mid-June. ERA-I shows better agreement with AMSR-E during this time period.

[Figure]

**Commented [DP23]:** Amended Figure 2 including ERA-I dataset segmented for freeze-up areas.

**Figure 2.** SnowModel hourly (solid lines), ERA-I daily (hashed lines) snow water equivalent (swe) accumulation and AMSR-E daily snow depth (crosses) converted to swe for freeze-up areas 1 (blue), 2 (green) and 3 (orange). The mean *in situ* swe and standard deviations for each area are displayed as circles at the end of November and colour coded to their respective freeze-up areas.

**Commented [DP24]:** Amended figure caption.

[Figure]

**Figure 3.** Mean (black), maximum (green) and minimum (orange) *in situ* measured snow water
equivalent (swe) for each site against mean SnowModel swe at each coincident model cell for the *in
situ* measurement period.

[Figure]

[Figure]

**Commented [DP25]:** New Figure 4 with amended swe scale

**Commented [DP26]:** New Figure 4 (b) displaying ERA-I snow accumulation grid.

**Figure 4.** SnowModel distribution map displayed as swe over McMurdo Sound, (a) fast ice, (b) open water/pack ice, (c) McMurdo Ice Shelf, (d) 
[revised manuscript text omitted]

**7 Conclusions**

This work has evaluated the ability of three independent techniques to provide snow depth on fast ice in the coastal Antarctic. SnowModel accurately captures the *in situ* measured snow distribution in November 2011 and produces a swe mean value that is 0.02 m above the mean of *in situ* validation, but when sea ice is segmented by fastening date large deviations of up to 5 cm are present in the east where the model has overestimated snow depth. This accurately captures the mechanism of snowfall and transport driven by the topography of Ross Island, but the rates are higher than in reality. ERA-I swe is 20 cm higher than *in situ* measurements and the gradient of the snow distribution produced by the analysis does not match is opposite to that measured *in situ*. A positive bias in accumulation should be expected from ERA-I as no snow redistribution mechanism is included. Any future work making use of precipitation reanalysis over Antarctic sea ice must include snow redistribution by wind, shown here by SnowModel to improve results. AMSR-E snow depth information suffers from problems already documented in the literature, and we find that its performance may have again been influenced by rough sea ice. The snow distribution produced by AMSR-E was opposite to that provided by SnowModel and measured *in situ* at the end of the growth season. We were unable to validate the instrument due to its failure 2 two months before the *in situ* data was collected. The uncertainty in the snow depth estimates manifest themselves in the sea ice thickness estimates from CS-2. The range in sea ice thickness uncertainty from the assumption that the snow surface or ice surface represents freeboard, as means of the entire growth season are 1.08 m, 4.94 m and 1.03 m for SnowModel, ERA-Interim and AMSR-E respectively. Using interpolated *in situ* snow information, Here, we find CS-2 freeboard measurements provided by the ESA retracker agree best with *in situ* measured thickness if are most likely representative of a mean dominant scattering horizon 0.07 m beneath the air-snow interface is assumed, in agreement with recent literature. It is impossible to confidentially constrain this number without reducing uncertainty in the established sea surface height from which the freeboard is estimated. This work demonstrates the need to reduce the uncertainty associated with the ambiguity of the altimeter radar freeboard measurement over Antarctic sea ice. Sea ice in McMurdo Sound is atypical of Antarctic pack ice, so improved understanding of the CS-2 freeboard measurement over varying snow and sea ice conditions in open water areas will be critical to accurately provide sea ice thickness estimates for the Southern Ocean.

Here, we show that modelled snow information has the potential to produce a time series of snow depth on Antarctic sea ice. However, major developments in modelling capability are required before their snow products can provide useful information for use in combination with altimetry data to provide sea ice thickness. Here, we show that modelled snow information has the potential to produce a time series of snow depth on Antarctic sea ice, that could be used with altimetry data to infer sea ice thickness if the reference surface of the altimeter can be accurately defined. 
[revised manuscript text omitted]

---

## Author Response (AR3)

**Response to final editor comments**

**The authors would like to thank the editor for the continuation of valuable comments on the manuscript. We have responded to the specific comments below in bold.**

Comments to the Author:

The authors have addressed all the remaining concerns of the reviewers. There correction to the ERA-I has substantially changed the results, however, the change further strengthens the main results that SnowModel performance is superior, and so addresses the comment that, based on the previous assessment, ERA-I did as well as SnowModel in matching the in situ snow depth. Therefore, I judge that the manuscript should be accepted. There are two points that the change in ERA-I has made more clear to me, and I would recommend that some short additional discussion be added to clarify these.

1. In the prior revision, I made the following comment:

"One thing you did not point out is that snowmodel takes as its input precipitation from Polar-WRF, which will be different from ERA-I. So the comparison between ERA-I and snowmodel and in situ (at east for the CS-2 comparison) mostly just shows that the retrieval is sensitive to errors in snow depth, and not which method is necessarily better (snowmodel would presumably be better where, as you note, snow redistribution matters, but you have not shown that this is a factor here)."

To which you responded:

It is stated in the manuscript (L244-248 second version, L251-255 latest version) that hourly atmospheric forcing were generated by version 3.5 of the polar-optimized version of the Advanced Research Weather Research and Forecasting Model. The revised ERA-I dataset also now show that ERA-I is not suitable for retrieving sea ice thickness in this region at least.

Due to poor wording on my part, this misses my point. The point is that the precipitation fields from WRF-ARW that are used to drive SnowModel are not that same as ERA-I. In fact, I believe Polar WRF initialization and boundary conditions are provided by NCEP GFS (this may have changed) so that it also cannot be viewed as a downscaled model of ERA-I. This means that ERA-I performing poorly does not suggest that SnowModel is responsible for the much better fits. It could be that the WRF-ARW precipitation fields forcing SnowModel are much better and responsible for most of the improvement. The results do suggest that snow redistribution in SnowModel may be important, as the snow depth distribution from SnowModel matches in situ distribution much better than ERA-I (figure 3), but it is not clear if the high-resolution model WRF-ARW uses in this area actually produces similar spatial variation in precipitation that ERA-I cannot, nor whether the match to mean snow depth is due to SnowModel or primarily due to better precipitation in WRF-ARW.

This was not necessarily a goal of the study, and not raised by the reviewers, but I would suggest at least that some comment in the discussion about this point and its significance for the results would be helpful.

**Thank you for pointing this out, it is an important point to clarify. We have decided to include an additional plot in Figure 4 (Figure 4a) to showcase the importance of the SnowModel redistribution by wind.**

[Figure]

**Figure 4.** (a) MicroMet swe distribution (b) SnowModel swe distribution in McMurdo Sound, (1) fast
ice, (2) open water/pack ice, (3) McMurdo Ice Shelf, (4) Ross Island. The model swe distribution is the
mean of the simulation over the *in situ* measurement period (25th November-1st December). The *in situ*
measurements were converted to swe via the density measured at each site, if no measurement was
taken (21 sites) the average *in situ* snow density was used (385 kgm-3). *In situ* measurement locations
are shown as black circles and are the mean of the 60 snow measurements taken at each site. The circle
sizes are weighted for swe to allow visualisation of the decreasing swe distribution from east to west.
Elevation contours are spaced at 400 m intervals; Mt Erebus (3,794 m) is the dominant topographic
feature on Ross Island to the east of the fast ice. (c) The interpolated 10 x 10 ERA-I grid with 1st
December accumulation total, the boundary of the SnowModel inset from (a) is shown as the black box.
The ERA-I centre points of the original grid are displayed as red dots.

We have also included the following in the text:

**Section 4:**

**"*It is important to note the importance of redistribution by wind which is provided by SnowModel. The consequences of neglecting this influence on snow accumulation in the study region are clearly demonstrated in Figure 4. Figure 4a displays the accumulated precipitation from MicroMet, while this is built on in Figure 4b with the inclusion of the other SnowModel components. Over eastern areas of the study region, the MicroMet precipitation output as a standalone product provides swe values double that of the highest swe measured in situ. Although vastly improved, the general overestimation of swe by SnowModel is clearly visible in Figure 4b.*"**

**Section 6:**

**"*Without this model component included the precipitation provided by MicroMet (downscaled PWRF) provides very poor estimates of snow depth on sea.*"**

In Figure 6, the cause of the range of the thicknesses is not clear, and not really discussed much in the manuscript. I can understand the range in ERA-I, since there is a large discrepancy here. But in the manuscript it is stated that SnowModel has a mean snow depth that is only 2 cm more (SWE, so ~5cm in snow depth) than in situ. Now, since equations 1-3 are all linear, I would have thought this offset ($dT_s$ ~ 5 cm) would cause an offset in the bounds shown by some modest amount (e.g. $dT_s*pw/(pw-pi)$, and $dT_s*(pw-ps)/(pw-pi)$ from the in situ bounds, or 30-50 cm). But in Figure 6 these changes are much greater (~ 1m). Also, since the Pd range shown would cover this ~5 cm bias, I do not understand why the range of the dots do not span the in situ depth indicated by the plus sign.

Either Figure 6 is showing something different, or I am not understanding how the bounds are being determined (is it showing the range of CS-2 derived thicknesses across the domain, and not the mean? Or possibly the 2 cm SWE bias is not scaled spatially, so the difference is mostly due to the larger bias in region 2?). It would be helpful to the reader for there to be a few sentences more explanation of what contributes to the differences in the results in Figure 6.

**Figure 6 displays the mean sea ice thickness (which is representative of all valid CS-2 retrievals in the study region) for each month of the sea ice growth season. The grey dots are the thickness assuming the air-snow interface is considered freeboard, the blue dots the thickness if the snow-ice interface is considered freeboard. The other linear fits between them represent the trends assuming different penetration factors. The large range in thickness through these different assumptions is because of the snow depth. ERA-I has the largest snow depth followed by SnowModel, followed by AMSR-E and the results reflect this. The bias is not scaled spatially and you are correct that the larger snow depths in regions 2 and 3 are driving the range in resultant thicknesses which also account for why the large penetration factors still do not meet the in situ thickness in Fig 5a. Fig 4b shows the high swe values which are in excess of 10 cm swe (30 cm + snow depth) for over half of area in fastening areas 2 and 3. In the very eastern areas values are as high as 100 cm of snow.**

**The forcing of this range is covered by the following "*From equations 1-3, sea ice thickness is highly sensitive to the snow-ice ratio for the measured freeboard. This results in a large range in sea ice thickness for all snow products through the growth season (Fig. 5). This range in inferred thickness is driven by the amount of snow produced by the models as Eq. 1 and Eq. 2 subtract and add the product of this value in their second terms respectively. As the snow depth increases, in some cases to higher values than the measured freeboard the Pd simply provides a correcting factor for this discrepancy.*" in section 5.**

**To provide clarity on the specific point mentioned above, the following sentences have been**
**included in section 5:**

**"*The ranges in sea ice thickness estimated with SnowModel as the snow depth input are substantially*
*smaller than ERA-I (Fig. 5), but still have a larger range than the mean discrepancy from in situ*
*measurements might suggest (Fig. 2). This is driven by CS-2 retrievals over the eastern areas of*
*fastening areas 2 and 3 where swe values are high, especially towards the end of the growth season*
*(Fig. 4b).*"**

Other minor comments:

Line 532-543, page 17. To clarify, ASPECT is from shipboard underway observations. These do not
necessarily avoid ridged areas, but the estimated snow depths are probably somewhat biased as they are
often estimated based on the snow depth on the level ice portion of the ice, and perhaps less from ridged
areas, and the ship likely does avoid heavily ridged, thick ice.

[revised manuscript text omitted]